# Marine Meteorological forecasts for Coastal Ocean Users - Perceptions, Usability and Uptake

Christo Rautenbach [a, b, c] and Berill Blair [d]

[a] *Coastal and Estuarine Processes, National Institute for Water and Atmospheric Research (NIWA), Hamilton, New Zealand*
[b] *Institute for Coastal and Marine Research, Nelson Mandela University, South Africa*
[c] *Department of Oceanography and Marine Research Institute, University of Cape Town, South Africa*
[d] *Environmental Policy Group, Wageningen University and Research, Wageningen, The Netherlands*

*Correspondence to*: Christo Rautenbach (Christo.Rautenbach@NIWA.co.nz)

**Abstract.** The present study aims to address a disconnect between science and the public in the form of a potential misalignment in the supply and demand of information known as the usability gap. In this case, we explore the salience of marine meteorological (metocean) information as perceived by users in two southern hemisphere countries: South Africa and New Zealand. Here, the focus is not only on the perceptions, usability and uptake of extreme event forecasts but rather focused on general, routine forecast engagement. The research was conducted by means of a survey, designed around three research questions. The research questions covered topics ranging from forecasting tool ergonomics, accuracy and consistency, usability, institutional reputation, and uncertainties related to climate change (to name but a few). The online questionnaire was widely distributed to include both recreational and commercial users. The study focused on identifying potential decision-making cultures that uniquely impact coastal ocean users' information needs. Cultural Consensus Analysis (CCA) was used to investigate shared understandings and variations in perceptions within the total group of respondents as well as in sectoral and country-based subgroups. We found varying degrees of consensus in the whole group (participants from both countries and all sectors combined) versus different subgroups of users. All participants taken together, exhibited an overall moderate cultural consensus regarding the issues presented, but with some variations in perspectives at the country-level, suggesting potential subcultures. Analysing national and sectoral subgroups separately, we found the most coherent cultural consensus in the South African users' cohort, with strong agreement regardless of sectoral affiliation. New Zealand's commercial users' cohort had the weakest agreement with all other subgroups. We discuss the implications from our findings on important factors in service uptake, and therefore on the production of salient forecasts. Several priorities for science-based forecasts in the future are also reflected on, considering anticipated climate change impacts. We conclude by proposing a conceptual diagram to highlight the important interplay between forecast product co-development and scientific accuracy/consistency.

*Keywords: Forecasting, Perception, Science communication, Survey, Cultural Consensus Analysis, Co-production*

## 1 Introduction

The accuracy of metocean predictions differ depending on the physical phenomena being forecasted. As an example, vertical ocean column structure parameters might be much more difficult to predict accurately than the prevailing ocean surface waves (in a very general sense as this statement is highly location dependent). The vertical water structure of both coastal and open oceans is driven by a larger number of environmental parameters which inevitably makes the physics, to be solved by numerical techniques, more challenging (including the requirement for 3D numerical considerations). This contrasts with 2D wave forecasts, which predominantly depend on local winds, offshore swell conditions and local bathymetry. Prediction techniques also play a large role in forecast accuracy, and have different computational demands associated with them. These include considerations of forecast time period, spatial extent and dimensionality, temporal resolution, and purpose. In the present study the perception, usage and uptake of metocean forecasts are assessed, predominantly focused on coastal and ocean winds and waves.

Around the world, operational centres clearly articulate the importance for user-centric (or transdisciplinary) based Research and Development (R&D) (e.g. Ebert et al. (2018)). Likewise, the broader climate services literature has focused on potential mismatches between the supply and demand of information that precipitates the so-called usability gap (Lemos et al., 2012; Kirschoff et al., 2013; Meadow et al., 2015; Zulkafli et al., 2017). Yet, limited anthropological studies have been conducted with user perceptions of science-based forecasts as the main research goal (Doswell, 2003; Silver, 2015) with the objective to gauge the extent to which groups of users do or do not share an understanding about what makes forecasts usable. Severe weather warning perception and uptake have been studied in the past (e.g. Sherman-Morris (2010)) but general (none-extreme) forecast usability, preferences and accuracy perception have not been extensively investigated (also known as the social aspects of weather or marine forecasting) (Silver, 2015). The few studies that did investigate the social aspects of weather forecasting include Demuth, Lazo, & Morss (2011), Katz & Lazo (2011), Lazo, Morss, & Demuth (2009) and Silver (2015). These studies are focused on North American countries (USA and Canada) and also illustrate how important weather forecasting is for economic development (Lazo et al., 2009).

Weather salience and the connection with atmospheric weather forecasts are discussed in studies by e.g. Stewart, Lazo, Morss, & Demuth (2012) and Williams, Miller, Black, & Knox (2017). The term 'weather salience' refers to the psychological importance weather has for a particular individual (Stewart, 2009). Several other studies started investigating how users' technical understanding and competence influence their interpretation and perception of

hydro-meteorological products (Ramos et al., 2010). Ramos et al. (2010) also encouraged users' technical training and direct engagement during operational forecast and hazard (early warning) tool development. This is especially true for probabilistic forecasting. Ramos et al. (2010) also highlighted the importance of exploring more effective ways of communicating forecasts.

User community perception is a crucial aspect of any marine-meteorological (metocean) information sharing or forecasting. Here the word forecast is used broadly to describe current and future earth system dynamics prediction. Several studies have established that active collaboration with users is needed to strengthen forecast service development, as a rich source of specific user interest and routines and as a framework for translating user needs into tractable research questions (e.g. Bremer et al. (2019); Lemos, Kirchhoff, & Ramprasad (2012); Meadow et al. (2015); Vaughan & Dessai (2014); Vaughan, Dessai, & Hewitt (2018); Wagner et al. (2020)). Codesign of services can help to provide the best information on relevant scales for all users and increase the rates of uptake. If user uptake or the enhancement of knowledge do not accompany the dissemination of forecast information, the forecast has limited relevance. Operational marine meteorological centres typically serve a wide range of clients with varying needs. The effectiveness with which relevant information is communicated to those clients can differ depending on the user's domain knowledge and the utilisation purpose (e.g. Kirchhoff et al., 2013; Lamers et al., 2018; O'Connor et al., 2005; Wagner et al., 2020). Specific clients often require bespoke solutions not entirely transferable to other users.

**1.1 Aim**

The present study aims to evaluate shared meanings of metocean forecast usability as important factors that drive the uptake of products, by engaging with members of the broader ocean community, with varying levels of ocean literacy and experience (e.g. recreational and commercial users). Confirming the knowledge viewpoints of these subgroups has not been investigated before and thus forms part of the present study. This research thus investigates the differences in the shared meanings of geographically separate groups: South African and New Zealand users. These two southern hemisphere countries are characterised by vastly different social structures and ocean states, and thus different social dynamics. Other than sharing the Southern Ocean and austral seasons, these countries both have heterogeneous ocean and coastal user communities. From a metocean perspective, they share similar climatologies and latitudes but on different continents with unique metocean dynamics.

Guiding research question include:

Q1: What important user requirements regarding usability impact marine forecast uptake by coastal ocean users in New Zealand and South Africa?

Q2: Will climate change affect the importance of those factors in the future?

Q3: Do geographic and sector-specific variations exist in levels of agreement pertaining to Q1 and Q2?

Questions 1 and 2 gauge present and anticipated future factors that impact forecast usability. The three questions together help us explore whether user perceptions regarding the usability of forecasting products are geographically/sectorally localized or if the two user groups share similar understandings of current and future forecasting needs. This was achieved by means of a questionnaire. By understanding users' points of view, metocean forecasting agencies/ companies can focus on providing relevant information in a format that enables effective uptake by better aligning the provision of information with its demand. This covers both commercial and public services such as commercial fishermen, search and rescue agencies, paddle craft clubs and surfers. The dual, southern hemisphere country investigation also provides a unique and relevant perspective on global, metocean forecast user needs. This is achieved through investigating two countries with extensive coastlines and diverse user communities.

## 2. Background

### 2.1 Perception, preference and uptake of forecasts

Silver (2015) investigated the perceptions, preferences, and usage of atmospheric forecasts information by the Canadian public. Environment Canada acknowledged the fact that their forecasts were reaching millions of citizens, but they were uncertain as to who or for what purpose these forecasts were being used. They thus investigated how their end users obtained, interpreted, and used their forecasts (Silver, 2015). They made use of both semi-structured interviews (n = 35) and close-ended questionnaires (n = 268). One of the most interesting findings from Silver (2015) was that forecasts were mainly used for pragmatic reasons. These would include checking the weather to decide what to wear for the day or for planning social activities, like going away for a weekend. The typical user did not pay attention to the ambient atmospheric conditions unless it was hard not to notice it (e.g. severe weather) (Silver, 2015). They also reported high levels of weather salience with regards to local weather knowledge. Most of the public were however unable to differentiate between products, e.g. what makes them different. The latter directly relates to understanding the basics of model forecasting horizons as well as spatial resolutions. Silver

(2015) also reported that the Canadian public trusted the Environment Canada weather forecasts and actively gave preference to their products. Silver (2015) highlighted numerous topics and questions that will be addressed and expanded upon in the present study, including the trust users have in various forecast products and why. This question is also even more interesting in the light of our changing climate. With the continuing rise in climate change impacts and changing weather patterns, user understanding, and uptake of forecast products have never been more important (a sentiment echoed in the results of the present study). Here, we will focus on ocean and coastal users and include marine forecasts as the main predictand.

In the Northern hemisphere, Finnis, Shewmake, Neis, & Telford (2019) presented a Canadian study where the marine forecasting needs of fishers were investigated and how the available marine forecasting products were used in their decision-making process. They followed a semi-structured interview process and found that there was a "subjective art" to the development/ dissemination and uptake of marine forecasts. Without a direct distinction between user groups, they found that forecasters (commercial/ specialist users) gave more attention to technical details, like model accuracy and consistency, while the fishers (commercial/ recreational) focused more on usability. Kuonen, Conway, & Strub (2019) also investigated the perception of risk associated with marine forecast products. Commercial fishermen were chosen as the main user group and their study highlighted how important user engagement is for successful marine forecasting. Once again, semi-structured interviews were used, and the study was based in the USA. These studies thus only had one user group as focus and did not consider a wider spectrum of typical ocean and coastal users. Other studies focused on forecast co-production in the northern hemisphere includes Bremer et al. (2019), Lemos et al. (2012), Lövbrand (2011) and Meadow et al. (2015).

A distinction may also be made between commercial users and the general public, the latter typically being a public good concern. The distinction between these user groups might explain some of the results observed by Silver (2015). The suspicion is that commercial, or specialist users, will display a higher level of understanding when it comes to technical aspects of forecast usability perception. Doksæter Sivle and Kolstø (2016) investigated the use of online weather information for everyday decision making. Here it became clear that this distinction is also dependent on the task (for which the forecast is used) and not only on the person or group. Marine information and forecast dissemination parameters include ocean winds, waves, temperature, current velocity, water level and water quality dynamics. Drift predictions, associated with search and rescue operations or oil spills, are examples of two services with major human and environmental consequences.

Limited studies have been performed linking southern hemisphere, metocean forecasting needs with available forecasting products. An example is presented by Vogel & O'Brien (2006) where they focused on the uptake of seasonal atmospheric forecasts over southern Africa. Hewitt (2020) also presented a high-level discussion on the challenges faced by the UK MetOffice in delivering climate services globally, including the southern hemisphere. The uptake of a metocean forecast depends on numerous factors beyond technical accuracy. Some are even related to the "look and feel" of the dissemination methods: e.g., are the forecasts being accessed via simple text messages, smart phone apps or via traditional publicly available media channels?

## 2.2 Geography, operational settings, and the cultural dimensions of ocean use

Most user perception related studies have been conducted in the northern hemisphere. Not only does the oceanography and atmospheric dynamics differ between hemispheres but so do the cultures established within this predominantly oceanic hemisphere. Both South Africa and New Zealand are in the southern hemisphere at similar latitudes. Both countries have a considerable coastline and are directly exposed to the Southern Ocean. South Africa used to be a crucial supply stop for ships traversing between the eastern and western trading routes (Worden, 2007) and currently has a coastline stretching approximately 3 000km. New Zealand, similarly, only has Australia as close by neighbour and is considered as being two islands with an approximate coastline of 15 000km. Due to their geographical locations, these extensive coastlines exhibit a variety of coastal, shelf scale and open ocean dynamics (e.g. Barnes & Rautenbach, 2020; Chiswell, Bostock, Sutton, & Williams, 2015; Godoi, Bryan, Stephens, & Gorman, 2017; Rautenbach, Daniels, de Vos, & Barnes, 2020).

The seafaring heritage of New Zealand resulted in a nation that tends to be interested and involved in everyday metocean predictions. A large portion of the country is aware of the ocean and technically everyone is near the ocean. This is also depicted in the traditional art of New Zealand (Dunn, 2003; Keith, 2007; Ministry for Culture and Heritage, 2014). The culture and language are also weaved into ocean-based references and symbolism (Wolcott and Macaskill, n.d.). One such example is the Mangopare (hammerhead shark symbol). The double Mangopare has been incorporated into the New Zealand MetService's logo and represents weather prediction and oceanography and their dependence on each other. This general stance was also reflected in the results presented in the present study. South Africa on the other hand has a much less direct relationship with the ocean. The European settlers were most directly linked with trading routes while the British came to colonise South Africa (Oliver and Oliver, 2017). South Africa is also part of the African continent, and thus the traditions and cultures

were much more terrestrial focused (Compton, 2011); the Khoisan people being some of the few with a true and
dependant relationship with the coastal oceans (Kim et al., 2014). Here Khoisan refers to the first indigenous
peoples of Southern Africa (Rito et al., 2013). Recently, South Africa made an active step towards focusing on the
ecosystem services (blue economy) their vast coastline can offer through a project called Operation Phakisa.
Phakisa roughly translates to "hurry up" in Sesotho (Findlay, 2018).

The type of relationship users cultivate with the ocean, and the resulting information need that is generated, is not
only driven by geographical contexts but also by sectoral differences that determine sociomaterial (linked human-
technological) settings (Blair et al., 2020; Lamers et al., 2018). Marine meteorological forecast users engage with
metocean information as a tool to mitigate risks. Attitudes toward risks are a result of a constellation of individual
and cultural factors, tied to bias, attitudes, preferences as well as societal influences and dominant worldviews
(Fischhoff et al., 1978; Douglas and Wildavsky, 1982; Lichtenstein and Slovic, 2006; Kahan et al., 2012). These
attitudes together can have a profound impact on the type of weather and climate information sought for decision
making (O'Connor et al.2005; Kirchhoff et al. 2013). We also know that mariners and the organizations underlying
navigation, develop distinctive traits based on unique mental models, organizations and decision-cultures (Lemire,
2015; Kuonen et al., 2019; Hederstrom n.d.) and these factors uniquely impact mariners' information needs (e.g.
Wagner et al., 2020). Forecast services are used in distinct ways in different sociomaterial settings, and these
differences impact the temporal and spatial scale at which information is needed for planning and tactical decisions.
Consequently, the socio-economic value that may be derived from salient forecasting services varies across a wide
spectrum of geographic and sectoral contexts as well.
As more interdisciplinary research includes diverse stakeholders and their observations about the technical, natural
and human factors that drive the need for information. It is increasingly apparent that understanding user needs,
often in cross-sectoral and cross-cultural settings, is a significant challenge. In this research we use the term culture
to denote learned ways of knowing; more specifically, learned knowledge that shapes people's approach to ocean
resources, and ocean information use. Culture affects users' perceptions about, and attitudes toward, technologies
in general (Lee et al., 2007; Lim & Park, 2013), and the meaning and relative importance of salient scientific
information (e.g. Martinson & Westwood, 1997). Traditional interview and questionnaire methods do not always
explain the variation in experiential knowledge that may exist across representatives of a wide range of sectors and
decision environments. We used Cultural Consensus Analysis (CCA) (Romney et al., 1986) to document this
variation and to look for patterns in user perceptions regarding the important factors that make forecast products
trusted and used.

## 3 Methods

CCA is a method that can reveal agreements among a group of people as a reflection of shared knowledge (Romney
et al. (1986)). Users' unique mental models, organizations and cultural domains result from specific practices and
operational contexts (refer to Section 2.2). Cultural consensus is an appropriate method to assess cultural domains;
in this case gauging the extent to which the practices and ocean use contexts of recreational marine users are of the
same cultural domain (i.e. they develop and share the same understandings about the factors that enhance forecast
usability) as professional users. CCA has been applied to study cultural populations and knowledge domains in
diverse fields, for example in public health (Garro, 1996; Weller et al., 2012; Strong & White, 2020), natural
resource management (Miller et al. (2004), Naves et al. (2015)), tourism studies (Paris et al. (2015), Ribeiro (2016)),
and studies of expert and lay knowledge (Medin et al. (2002), Reyes-Garcia et al (2006), Van Holt (2016)).
This study contributes to knowledge about human dimensions such as cultural values and understandings that
influence the direction of forecast products and services development. The consensus model can show shared
understandings among users of forecasts to reveal patterns of understanding and meaning that impact the adoption
of services and products. An advantage of cultural consensus analysis is that a small population of respondents can
yield rich observations and data regarding sector (commercial and recreational) or locality-specific (South African
and New Zealand) views and knowledge-domains as they may exist among participants (Weller (2007)). The
present study aimed to test the knowledge domain differences between New Zealand and South African user groups
(as well as recreational versus commercial users) toward what constitutes salient forecast service. There is a
common perception that there does exist a difference between these user groups, but no formal investigation has
yet been done to confirm these suspicions.

### 3.1 Questionnaire

In this study, recreational users include all participants who do not use metocean forecasts as part of their daily
work or do not have a financial gain from the use of such platforms. Commercial users would then automatically
be the other users, who not only use the platform commercially but also have responsibility linked with the
understanding and accuracy of these forecasts. The questionnaire asked the participant to identify themselves within

one of these definitions. The questionnaire was organized around four sub-questions linking to our research questions (Q1 and Q2 in Section 1.1):

1.      *Which factors impact marine forecast uptake by marine users?*

2.      *What are the main requirements from users in the marine forecast environment?*

3.      *What is the user perception of existing wave forecasting platforms?*

4.      *How important will accurate metocean forecasts be in the future (in light of climate change)?*

The questionnaire presented propositions in true/false format developed around a diverse collection of 27 constructs. The constructs were selected in a workshop with experts in the metocean forecast industry, based on issues that had frequently emerged in dealings with users in the past. The workshop members were from the meteorological service of New Zealand and the South African Weather Service (SAWS). Contributing scientists' competencies spanned atmospheric, hydrodynamic and wave forecasting and observations. Some scientists also had experience in science communication and client liaison and familiarity with the decision space (or operational context) of their respective user groups. The resulting propositions regarding these constructs, per research question, were then collected and refined.

The questionnaire was widely distributed. The questionnaire was advertised to both recreational and commercial users throughout both countries (New Zealand and South Africa). Coastal and ocean users emailing lists and websites were used to spread the invitation as well as personal contacts. It is important to note that no ethical issues were encountered during the present study. No personal, identifiable information was collected during the survey. The identities of the participants are unknown, even to the authors, and thus fully anonymised. No institutional nor funding agency ethical clearance was required.

**3.2 Data Analysis**

The consensus model (Romney et al., 1986) estimates shared beliefs relying on three basic steps. First, it uses Principal Component Analysis (PCA) to test whether the responses are consistent with an underlying shared model for the topics covered in the survey. Eigenvalues are calculated to find a shared knowledge-domain, determined by the presence of a single factor that explains most of the variation in the responses, with a first to second eigenvalue

ratio greater than, or equal to, 3.0. Secondly, the model provides a measure of individual knowledge for each respondent (a type of 'competence' in the specific shared mental model) by testing each respondent's agreement with shared beliefs via a proportion match matrix that has been corrected for guessing. And finally, it aggregates individual answers to questions by weighting the final cultural model in favour of respondents with high competence. This set of responses produces the consensus-based result, an approximation of the collective knowledge of the group. The minimum sample size required for the consensus model depends on the level of agreement, the number of informants, and the validity of the aggregated responses (Weller, 2007). For example, at a low-level agreement of 50% (mean competence score of .5) at .95 validity, the minimum sample size is twenty-eight people per group. The same at 60% agreement is seventeen people. For data analysis the present study used the match coefficient method, of the formal consensus model, in the UCINET software package (Borgatti, S.P., Everett, M.G., and Freeman, 2002).

Cultural consensus analysis uses 'cultural competence' in very context-specific ways. Culture refers to shared sets of learned knowledge and beliefs among a group of people. Competence is the individual's level of expertise with regard to the set of questions presented, indicating the proportion of items each person knows about the particular domain without moral judgment (Weller, 2007). Similarly, the method identifies the 'culturally correct answers' to propositions, from consensus-based results or the most frequently held items of knowledge and belief.

## 4 Results

### 4.1 Participant demographics

In total there were 157 respondents to the questionnaire. New Zealand received 126 completed responses and South Africa received 31. These numbers proved to be sufficient for the use of CCA because the level of agreement (mean competence scores >= 0.5) and eigen value ratios (> 3.0) obtained in all cohorts (New Zealand, South Africa, commercial, recreational users) were above the required twenty-eight people per group (refer to Table 1). It was possible to establish consensus models despite the different participation rates and small sample sizes because in CCA validity is a function of level of agreement (Weller, 2007). A demographics related section was added as a part of the questionnaire. This enabled the present study to have insights into some crucial information that could explain trends observed in the CCA. These results are given in Figure 1, 2 and Appendix 1. The questions are listed from A to G together with the total responses.

In New Zealand most respondents classified themselves as recreational users (~84%). South Africa had a similar
result but with a much larger percentage of respondents being commercial users (~42%) versus the majority
recreational users (~57%). These results are particularly interesting given the next set of questions (refer to
Appendix 1, Questions B and C). In New Zealand, most of the respondents did in fact have both theoretical and
practical ocean/ maritime related training (~70% and 68% respectively). Even more so in South Africa, with ~73%
and 82% of respondents receiving theoretical and practical training respectively. Thus, it is not only individuals
engaging with the ocean in a professional manner that received ocean related training at some point in their lives.
This could also mean that even though people work in an ocean related industry (technically commercial users),
their relationship with metocean forecasts are for recreational purposes. There thus might also exist a disconnect
between metocean forecasts used professionally (possibly from other specialised, commercial providers and not
the same tools used recreationally) versus freely available tools, platforms and products. These thoughts then lead
to the next section of questions related to metocean forecasting platform usage and experience (refer to Figure 1,
Questions D and E).

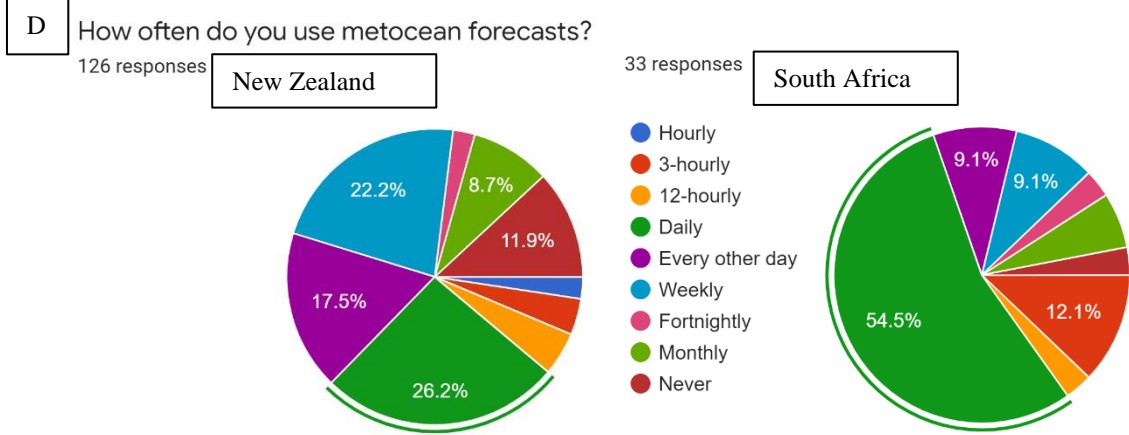


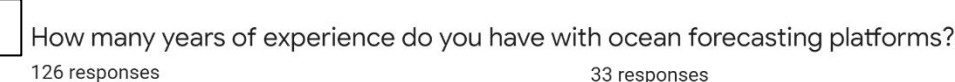

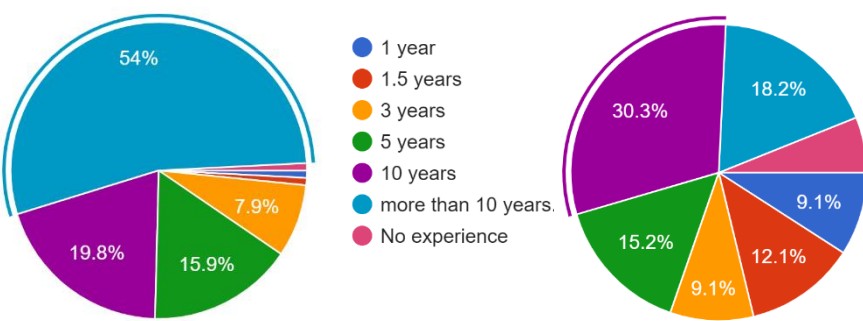


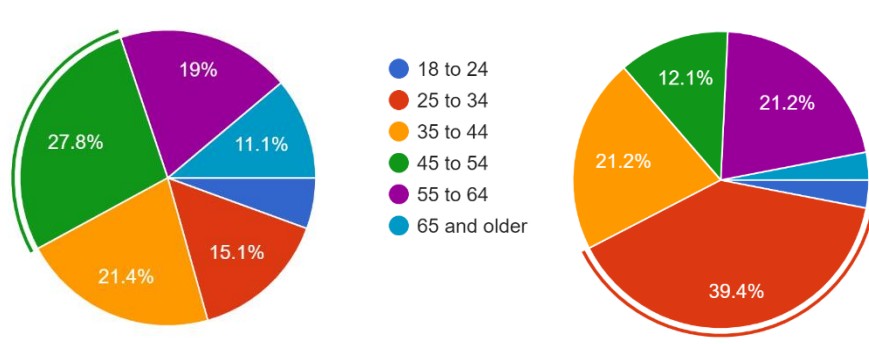



**Figure 1:** Summary of demographic questions related to the present study. Here Questions D to F are given with Questions A to C given in Appendix 1, together with their results.


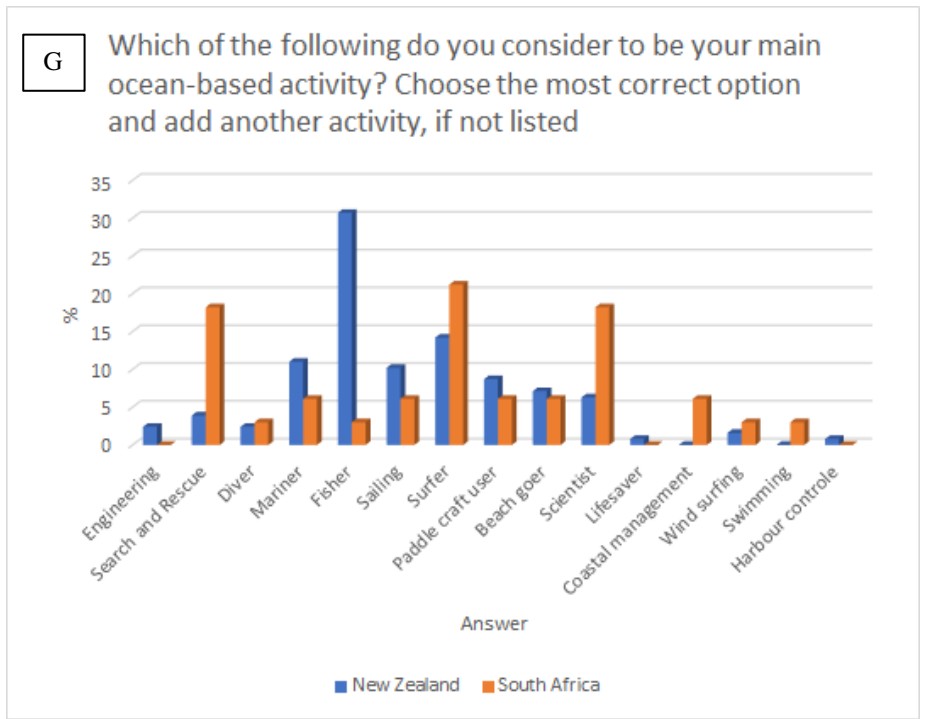

**Figure 2**: Summary of demographic questions related to Question G.

In New Zealand the most popular frequency of use ranged between daily, weekly and every other day (~ 26%, 22% and 18% respectively). In South Africa most of the usage was daily (~55%), then 3-hourly (~12%) and every other day (~9%). From these results it seems that most people will only look at a forecast once a day, probably for planning purposes. This agrees with the finding of Silver (2015), where they found that people might consult a forecasting service once during the planning of an outdoors activity. In the context of this study, it will be an ocean and coastal related activity. While South African participants consult forecasts at a higher frequency, New Zealand participants had much more experience compared to the South African respondents. ~54% of New Zealand respondents had over 10 years' experience using metocean forecasting platforms. ~ 20% had 10 years' experience (refer to Figure 1, Question E). In South Africa the majority of respondents had 10 years' experience (~30%) with ~18% more than 10 years' experience. In general, South Africa had more diversity in age with a larger contingent with less than 3-years' experience. These results correspond to the age of participants in Figure 1, Question F. In New Zealand most respondents were between 45 and 54 years old while in South Africa the majority were between 25 and 34 years old. Both countries have a significant contribution from the age brackets between 35-44 and 55-66 with New Zealand also having a significant number of participants older than 65.

In Figure 2, Question G was related to the actual activities respondents (recreational and commercial) engaged in. Participants were also given the opportunity to add activities that were potentially not listed in the questionnaire. The only two activities that stood out as not being listed, and thus recommended by a few respondents, were water-skiing and photography. In New Zealand most respondents use the ocean for fishing activities (31%) while in South Africa most respondents were surfers (~21%). The other significant New Zealand activities were surfing (~14%), mariners (~11%) and paddle craft users (~9%). The other prominent South African activities were Search and Rescue operations (~18%) and scientific studies (~18%). The questionnaire also asked how many years' experience each respondent had in ocean related activities (these are activities and not the use of forecasting platforms indicated in Figure 1, Question E). For the New Zealand users, 81% indicated more than 10-years' experience while South Africa revealed ~60% with more than 10 years' experience, ~18% with 10 years' experience and ~12% with 5-years' experience. For both countries very few respondents had less than 3-years' experience in ocean related activities. In Figure 2 the participant distribution in both New Zealand and South Africa is provided.

In Figure 3 the participant distribution in both South Africa and New Zealand is given. As a final note on the geographical context, ~50% of New Zealand respondents were from the Auckland district, ~16% from the Waikato, ~11% from Wellington and ~10% from Northland. Representation was also received from the other districts (both on the North and South Island). In South Africa most respondents were from the Western Cape province. More specifically, ~49% from Table Bay and the Atlantic Seaboard, ~15% from Kommetjie- Cape Point and ~9% from Simons Town in False Bay (also the location of the South African Navy headquarters). Very few to no participation was received from the eastern provinces of South Africa.

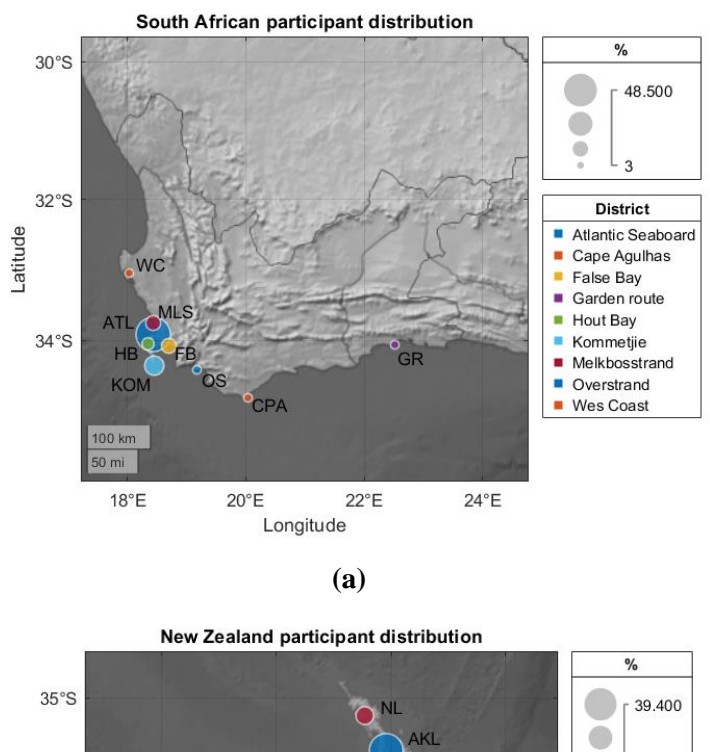

**(a)**

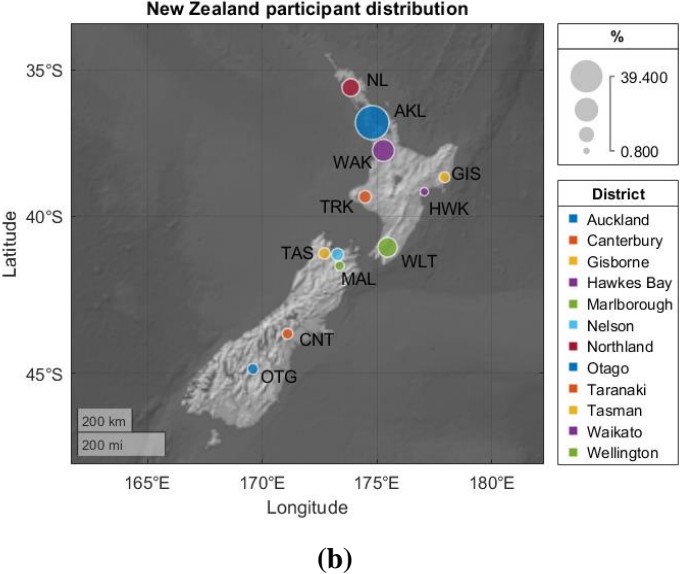

**(b)**

**Figure 3:** (a) New Zealand and (b) South African participant distribution.

It should be mentioned that the participants were also questioned regarding their trust in, and perceptions of, their own national weather services. In South Africa it is the South African Weather Service (SAWS) and in New Zealand the MetService. The greatly diverging perceptions in the two groups, regarding their own national weather services provider, may present pre-existing biases that would have to be addressed subsequently in the consensus

analysis. These questions were regarding the meaning of salient services. However, both institutes were evaluated
very highly and were found to be trustworthy (agreement: NZ 75%, SA 61%), reputable (NZ 77%, SA 58%), high
quality (NZ 68%, SA 84%) and reliable (NZ 71%, SA 74%). 58% of New Zealand participants agreed that their
national weather service produces marine products with likeable visual appeal, while 49% of South African
participants said the same about their respective service.

## 4.2 CCA results

### 4.2.1 Degrees and patterns of consensus among respondent groups

We found that respondents in both countries and in both user-type groups displayed an overall similar answer
pattern, and the data indicated broad agreement about the propositions presented in the survey. As indicated in
Table 1, for all scopes of analysis (see five consensus models in column 1) the ratio between the first and second
eigenvalues was above the 3 to 1 ratio, suggesting that there was a shared mental model regarding the main factors
that impact user uptake of metocean forecasts. Analysis of the entire dataset consisting of all respondents and their
responses to each proposition (whole-group model), resulted in an eigenvalue ratio of 6.34 (subgroup model
eigenvalue ratios ranged from 4.82-8.04). This finding suggests that respondents across all geographic and sectoral
contexts share some of the basic understandings about what constitutes salient marine forecasts.

The present study found varying degrees of consensus in all five consensus analysis runs conducted. Separate
consensus analyses among subgroups from different communities and sectors displayed slightly varying answer
patterns (refer to Table 2) and levels of agreement. For a detailed writeup of noteworthy variations in Table 1 the
reader is referred to Appendix A. Analysis showed the average estimated competence score of the respondents to
be 0.53 (SD = 0.17) in the whole-group consensus analysis (South African cohort: 0.61; New Zealand Cohort: 0.51)
(refer to Table 1). The eigenvalue ratio and average estimated competence scores at first glance indicated that
despite regional differences in geophysical conditions and sectoral differences in sociometrical contexts, marine
users generally agreed about important requirements for marine forecasts. But there was high variability in mean
competence scores in some of the consensus models. We adopt the heuristic by (Caulkins and Hyatt, 1999) to help
distinguish varying degrees of consensus, where multiple centers of agreement may exist and form so-called
noncoherent models. Where multiple negative competence scores exist, and/or where one subgroup's mean
competence is less than .5 (while the other is significantly higher) we identify the model as noncoherent regardless
of the eigenvalue ratio. Negative competencies would signal that a participant responded very differently from
others.

**Table 1:** Cultural consensus analysis, group mean competence scores and eigenvalue ratios of the first to second
factors for each study region and sector. An individual's competence score is the probability that the informant
knows (not guesses) the answer to a question, and it is a value between 0 and 1. A group's average estimated
competence score above 0.5 indicates moderate agreement in the group, pointing to an underlying model of shared
knowledge. Five consensus models were calculated (column 1), for each consensus model the breakdown of mean
competence scores along group membership is shown for comparison. Conclusions regarding the consensus model
are based on criteria by Caulkins & Hyatt (1999). Here, SD refers to the Standard Deviation.

| Scope of analysis: | Eigen value ratio | Mean competence score (SD) | Mean competence score (SD): South Africa | Mean competence score (SD): New Zealand | Mean competence score (SD): Commercial users | Mean competence score (SD): Recreational users | Negative competence scores | Conclusions |
|---|---|---|---|---|---|---|---|---|
| **Whole group consensus model (all respondents) N = 157** | 6.34 | 0.53 (0.17) | 0.61* (0.12) | 0.51 (0.18) | 0.53 (0.19) | 0.53 (0.17) | 1 | Coherent model: moderate agreement |
| **South Africa consensus model N = 31** | 8.04 | 0.61 (0.12) | - | - | 0.6 (0.12) | 0.6 (0.13) | 0 | Coherent model: strong agreement |
| **New Zealand consensus model N = 126** | 5.36 | 0.50 (0.18) | - | - | 0.45 (0.20) | 0.51 (0.17) | 3 | Non-coherent model: multicentric, contested |
| **Commercial users' consensus model N = 34** | 4.82 | 0.52 (0.21) | 0.62* (0.12) | 0.44 (0.23) | - | - | 1 | Non-coherent model: weak agreement |

| Recreational users' consensus model N = 123 | 6.4 | 0.53 (0.17) | 0.62* (0.13) | 0.52 (0.17) | - | - | 1 | Coherent model: moderate agreement |

*significant at p < .05


The presence of agreement among the group as a whole (and within each subgroup) was checked, visually, with
multidimensional scaling (MDS) (refer to Figure 4 for whole group agreement and Figure 5 for all subgroups)
which confirmed overlapping agreement among subgroups with some scattering of low competence score
participants. These visualizations depict the proportion of similarities between respondents' answer patterns in a
scatter plot. The x and y axes do not represent meaningful numeric values beyond communicating relative distance
between objects. Those who had high levels of agreement with each other are situated close to each other, while
those who had high levels of disagreement are scattered proportionally farther apart. The blue oval gives an
approximate grouping of all respondents who had a competence score of 0.6 or higher. The stress value is the
distortion that occurs when data are transposed over multiple dimensions. These values are reported in the figure
captions and in all cases meet criteria set by Sturrock and Rocha (2000).

The whole-group consensus model (refer to Figure 4) indicates that most South African respondents (red squares)
cluster closer and centrally located together with New Zealand respondents (blue squares) who have high individual
competence scores. This group, at the centre of the plot, had the highest levels of agreement with other respondents
and therefore the highest competence scores. Respondents who are more peripheral and scattered outside the blue
zone had lower competence scores: the farther away their location, the lower their score. These individuals
frequently answered propositions differently than the consensus model. Peripheral individuals located on opposite
sides of the plot had high levels of disagreement not only with the consensus model but also with each other. South
African respondents who are outside of the blue zone are still located relatively close to the centre, compared with
the outliers farthest away who belong in the New Zealand subgroup. New Zealand commercial users are
disproportionately represented on the outside of the blue oval (13 of 20 individuals) in Figure 4, aligning with
findings based on patterns of agreement and mean competence scores (Table 1) in the various subgroups.

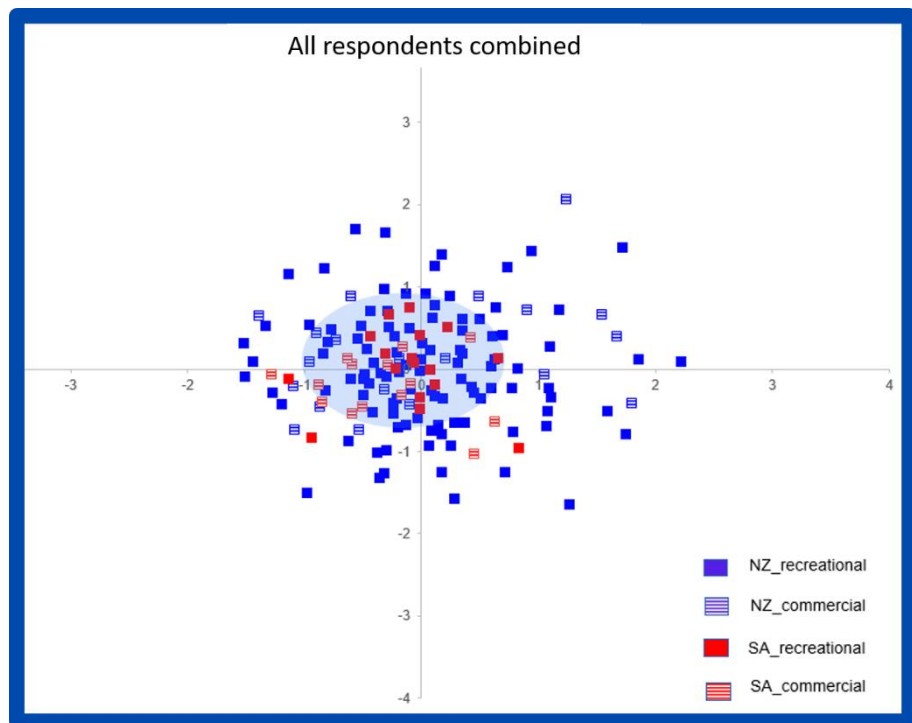

**Figure 4:** Nonmetric, multidimensional scaling of agreement in the whole-group analysis (stress = 0.264). Blue oval at centre is an approximate grouping of respondents whose competence score was 0.6 or greater.

Country/sector-specific and community-specific analyses revealed that commercial users from New Zealand have unique patterns of agreement, independent of whether the analysis includes fellow New Zealand users such as in the New Zealand consensus model with mixed sectors (Figure 5 (A)), or South African users in the commercial users model with mixed geographies (Figure 5 (D)). The visualizations indicate that commercial users from New Zealand scatter outside the blue oval in disproportionate numbers. Commercial and recreational users from South Africa demonstrated equally high levels of competence in their shared consensus model (Figure 5 (B)). When the South African commercial and recreational user groups were analysed in sector-specific contexts with their New Zealand counterparts (commercial and recreational users consensus models), both groups demonstrated significantly higher shared competence scores than New Zealand participants (see also Figure 5 (C)). This means that South African respondents have a more homogenous shared mental model among themselves and they share high levels of agreement with New Zealand users who attained high competence scores. Further studies are needed that explore the knowledge domain of New Zealand commercial users, with regards to forecast needs and perceptions about existing services. In this study the number of participants in this cohort was too low for a separate

consensus analysis. For now, the conclusion is made that this cohort's understanding on the issues did not conform
well to that of other cohorts (refer to Table 1).

In the next section we present the answers (the consensus results) in each group of analysis, for a comparative
analysis of the ways in which locality (national affiliation) and sectoral affiliation resulted in the same or different
answers to our questions.

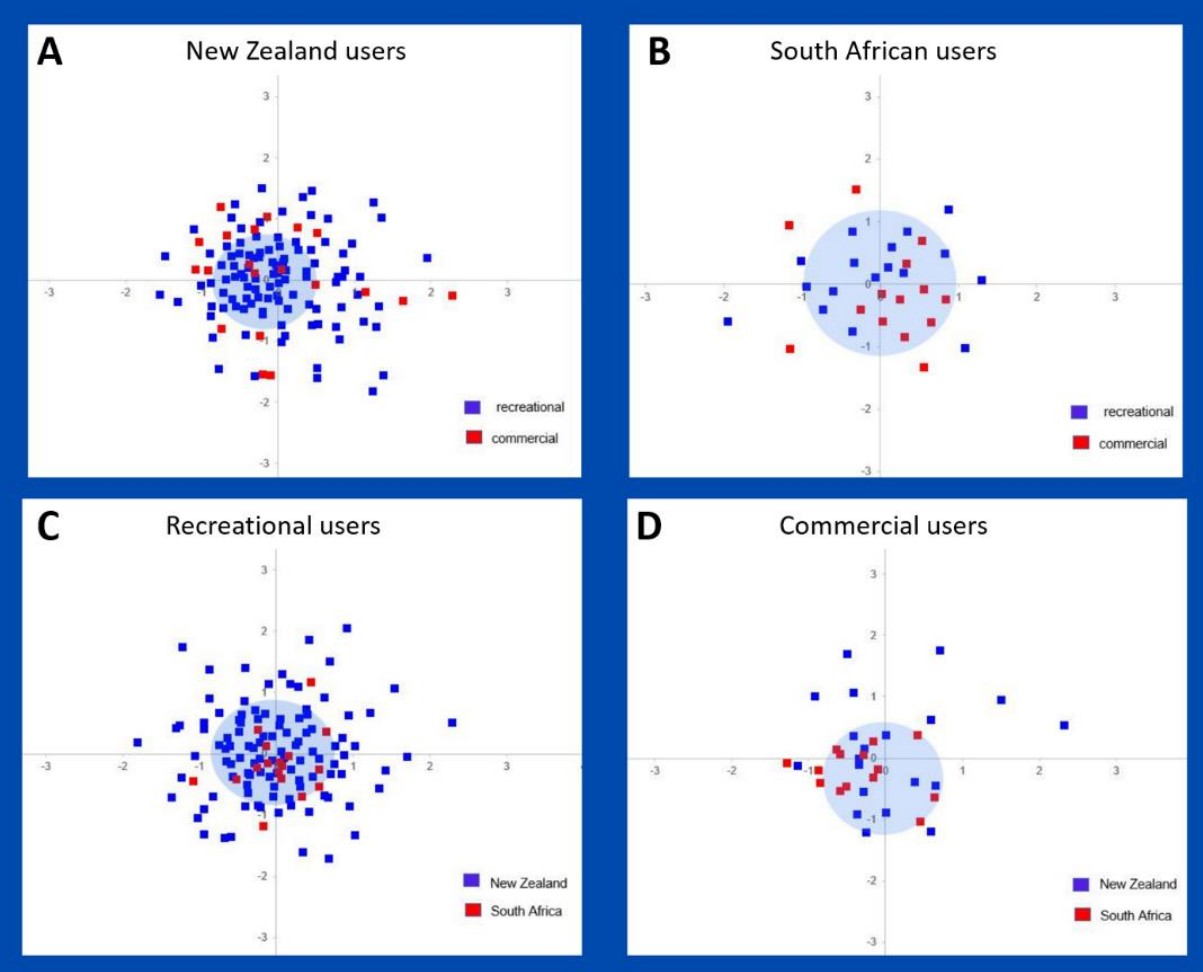


**Figure 5:** Nonmetric, multidimensional scaling of agreement in the subgroups. Blue oval at centre is an
approximate grouping of respondents whose individual competence score was 0.6 or greater. Panel A: New Zealand
ocean users (stress = 0.263); Panel B: South African ocean users (stress = 0.237); Panel C: recreational ocean users
(stress = 0.258); Panel D: commercial ocean users (stress = 0.207).

**4.2.2 The consensus model: factors that impact user uptake of metocean services**

Table 2 presents the results of the survey. These are the direct questions and resulting propositions that were distributed in the survey and form the basis of the present study. The column titled "whole-group CCA" is based on the consensus analysis of all respondents together, and it shows the aggregate group belief (culturally-correct answer) with either agreement (green icon) or disagreement (red icon) with the propositions. The other columns indicate the percent frequency of matching answers (or agreement with the whole-group CCA), in each subgroup. In case a subgroup's own consensus-model (consensus analysis run only including its members) produced a group belief that deviates from the whole-group CCA, the added icon indicates the correct answer in the sub-group.

**Table 2: Level of consensus measured by the frequency of culturally correct answers (CCA) for all propositions. The whole-group CCA is based on the analysis of the entire dataset consisting of all respondents; the culturally correct answer set (consensus model) is shown as either true/agreement (with a green icon) or false/disagreement (with a red icon). Numeric values are percent of responses matching the whole-group CCA in the relevant subgroups. Where a subgroup's own consensus-model (consensus analysis run separately only with members) deviates from the whole-group CCA, the added icon shows the correct answer in the sub-group.**

| Topic areas | Research questions and propositions | Whole-group CCA | NZ subgroup | SA subgroup | Recreational users subgroup | Commercial users subgroup |
|---|---|---|---|---|---|---|
| | **Which factors impact marine forecast uptake by marine users?** | | | | | |
| Ease of use | The visual experience offered by a forecast | ✅ | 84 | 90 | 85 | 88 |
| Easily cross-referenced geographical parameters | Easy access to location of interest | ✅ | 64 | 55 | 63 | 59 |
| Number of clicks | Number of clicks to relevant information (less is better) | ✅ | 81 | 84 | 80 | 85 |
| Easily cross-referenced physical parameters | Easy access to variable of interest | ✅ | 77 | 84 | 77 | 82 |

| Category | Description | | V1 | V2 | V3 | V4 | V5 |
|---|---|---|---|---|---|---|---|
| Institutional reputation | Whether provider is an established entity or a "newcomer"* | | ✗ | 56 | ✓ | 55 | ✓ |
| Terminology | Use of jargon or scientific terminology makes a forecasting site: | Intimidating** | ✓ | 56 | 84 | 59 | 71 |
| | | Untrustworthy | ✗ | 90 | 90 | 90 | 91 |
| Marketing | Word of mouth and recommendation by peers | | ✓ | 90 | 97 | 92 | 88 |
| Accuracy | Inaccurate forecasts (loss of trust in provider) | | ✓ | 74 | 71 | 74 | 71 |
| Consistency | The consistency of inaccuracies (forecast can still be useful if consistent)*** | | ✓ | 66 | 74 | 67 | 68 |
| Community engagement | Interactive features (ability to submit photos, info is better) | | ✓ | 48 | 61 | 49 | 56 |
| Simple metrics | Simplified concepts, graphs and plots and easy-to-understand, quick uptake scaling of metocean conditions | | ✓ | 70 | 74 | 72 | 65 |
| Intuition / experience | User's own intuition as a part of the safety calculus/decision making when predicting conditions | | ✓ | 73 | 84 | 77 | 68 |
| | **What are important requirements from users in the marine forecast environment?** | | | | | | |
| Speedy answers | The length of time taken between navigating to a forecast service and arriving at the desired data | | ✓ | 86 | 100 | 87 | 94 |

| | | | | | | |
|---|---|---|---|---|---|---|
| Bespoke forecast | Customizable preferences to facilitate faster access to desired information | ✅ | 93 | 94 | 92 | 97 |
| Forecasting horizon | A forecasting period between 3 and 7 days | ✅ | 92 | 97 | 93 | 91 |
| Training | Training in the science behind and use of marine forecasts | ✅ | 56 | 65 | 55 | 65 |
| **What is the user perception of existing wave forecasting platforms?** | | | | | | |
| Public platforms (e.g. Windy, Windguru, Magicseaweed and Buoy Weather): | have a high reputation among marine users | ✅ | 85 | 94 | 87 | 85 |
| | are reliable for most locations in the nearshore | ✅ | 65 | 61 | 71 | ❌ |
| | are most useful further away from the coastline | ❌ | 61 | 48 | 64 | ✅ |
| | have a likeable visual appeal | ✅ | 87 | 94 | 87 | 94 |
| **How important will accurate metocean forecasts be in the future?** | | | | | | |
| Reliability | Reliable metocean forecasts will be even more important | ✅ | 82 | 94 | 85 | 79 |
| Consequences | The consequences of mispredictions will be more severe | ✅ | 73 | 65 | 72 | 68 |
| Climate change | Climate change is making the ocean more difficult to predict | ✅ | 48 | 68 | 51 | 53 |
| Institutional reputation | The scientific reputation of forecast providers will become more important | ✅ | 75 | 81 | 76 | 79 |
| Scientific support | Science based forecasts will be more important in the future. | ✅ | 87 | 90 | 87 | 88 |
| Training | Climate change will make an understanding of the science behind ocean forecasts more important | ✅ | 75 | 100 | 78 | 88 |


*Respondents suggested that while familiarity and established trust in a provider can encourage uptake of services, users are open to
newcomers and view some of their products as very trustworthy.
**The New Zealand and recreational users' subgroups indicated that users are generally able to figure out the meaning of technical
terminologies.

*** Respondents noted that while in such cases the forecast can still be useful, the inaccuracies decrease usefulness.

The first research question explored which factors impact marine forecast uptake by marine users. These factors
range from aesthetics to practical considerations, like the number of clicks required to get to the required
information. All users and regions rate the ease of use as being very important. This includes easy navigation and
ergonomics of the tool or site. The opinion of others is also important to all users. So, if a site is being promoted
through a community via word of mouth, uptake and usage of the forecasting site or tool will increase. It is also
interesting to note that if a forecast is inaccurate, there is a significant proportion of the user communities that
would not necessarily stop using the forecast, as long as the inaccuracies are consistent.  The South African and
commercial users' subgroups agreed that services from established entities are trusted more than those offered by
newcomers, while all subgroups agreed that intuition (in combination with forecast products) helps to keep
operations safe.
When considering the requirements from users, speedy answers were strongly agreed upon, so much so that 100%
of South African respondents, regardless of sectoral affiliation, agreed. All users agreed on a preferred forecast
horizon (3-7 days) and that training on the use of products is needed. The conviction about training was not as
strong as the other propositions, with the sentiment strongest expressed by all South African users and the
commercial user's subgroup.  Well-known wave forecasting platforms are trusted and enjoyed by all user groups,
but perceptions about the location of highest accuracy varied. The fourth and final research question is related to
climate change and the uncertainties associated with it. All groups and subgroups agreed that reliability of metocean
forecast will be more important in the future and the role of training in forecast use will be even more significant
for safe operations. Consensus was weak however, around an overall agreement, that climate change impacts will
make the ocean more difficult to predict.
**5. Discussion**
The results presented in Section 4 elucidated numerous interesting behaviours within regional (or sector) groups as
well as community groups. Part of the aims of the present study was to explore the existence of a common or global
typology for salient forecast services that spans geographic and sectoral contexts, to the extent it is possible. In
doing so, we also aimed to establish subgroup-level perceptions that are unique to specific contexts among
metocean forecast users. Using two southern hemisphere countries as test cases, some shared fundamental factors

in salient forecasts, and context-specific distinctions were thus confirmed. Numerous studies acknowledge varying user needs and opinions but the delineation between recreational and commercial users has not been suggested or illustrated before. Understanding user needs are very well understood in other commercial industries, but in the everyday metocean forecasts the connection between research, products and user needs are not well established. Even more so in the southern hemisphere, in every-day (none-extreme), forecasting domains. Drawing the results together into a clear discussion requires the consideration of all the results, including the demographic description provided in Section 4.1. The discussion will follow the results presented in Table 2 and draw on all the other results to elucidate user perceptions, usability, and uptake.

Another interesting outcome was the user relationship with the organisation or institution providing the forecast. In the past, users knew of state-owned research institutes with well-established reputations. This instilled trust from the users without much question. When new and unknown companies brought new products (especially science related) to the market, users were sceptical (Li et al., 2008). Through the development of technology, the public has grown accustomed to providers that they have never heard of before. Apps, websites and online shopping have changed the way society sees the world and inevitably their trust relationship with tools, products and services. This is reflected in the survey results, where the total CCA knowledge model disagreed on whether an institution is established or not matters much. The South African and commercial users' subgroups did however agree with this statement, aligning with findings from an investigation of the trust in Environment Canada's forecasting products (Silver, 2015). Therefore, evidence suggests that commercial users do still require institutional reputation, probably because there will be consequences for them based on the reliability of the forecast. Scientific integrity will continue to be an important factor in users' trust in products and services, and therefore, in their uptake.

All user subgroups confirmed that their own intuition plays an important role in predicting conditions and safe operations. The demographics presented in Section 4.1 supports this, as a significant number of users had a lot of experience with coastal and ocean activities and with metocean forecasting platforms. Consistently inaccurate forecasts were also mainly perceived as being useful. This also testifies of more experienced users as they will be able to recognise recurring inaccuracies and knowingly compensate for these. For example, if a significant wave height forecast for a particular region is always underpredicted, the users (through experience) can compensate for it. If the inaccuracy is erratic, this becomes impossible. The recreational surfing community is a good example of a community that applies local knowledge daily to compensate for model and forecast inaccuracies. This community tends to be expert metocean forecast users and have learnt how to interpret particular synoptic scale

events and forecast to sufficient accuracies of metocean conditions in the nearshore. Their interpolation (of wave conditions from the offshore to the nearshore) also exceeds most high-resolution models and (mostly) unknowingly compensate for various coastal processes (like friction, refraction, shoaling etc.). The same reasoning applies to most commercial users (including Search and Rescue operators).

The importance of a bespoke forecast was highlighted by very high levels of agreement (>90%) among respondents. This aspect of forecast delivery is still underexplored by numerous metocean forecast providers and thus requires investigation and further development. A three to seven day forecast horizon seemed to be preferable for most users. Much like the farming community, there still exists the need for longer term and seasonal scale forecasts as well. These are predominantly used for planning purposes by aquaculture farmers, coastal hazard assessments and governance authorities (Alexander et al., 2020). But for most users, who also use metocean forecasts daily (refer to Section 4.1.) short-term forecasts are most useful, probably due to pragmatic activity planning purposes (Silver (2015)).

Well-known metocean forecasting platforms were well-reviewed on reputation and visual appeal. These platforms do not necessarily conduct independent research on model calibration, validation, or improvements in the underlying physics. They generally repackage freely available forecast products in an easy to understand and ergonomic fashion. The features of most of these sites are user-centrically designed and thus enjoy high esteem from all users (as confirmed by the present study as well). Most of these repackaged, freely available products are not accurate or reliable in the nearshore. This is due to model resolution and the presence of land. Both atmospheric and oceanographic parameters do not take nearshore topography or bathymetry into account and can thus not solve the relevant physics with high enough detail. The degree to which these models are inaccurate will vary depending on the coastal location. The commercial users' subgroup CCA model was the only cohort that disagreed with the proposition that these models/ platforms are reliable in the nearshore. This is an indication that commercial users are more aware of the underlying assumptions of these models. This is also reflected in the South African cohort, as their commercial representation was larger (refer to Section 4.1). These models are in fact more useful and accurate further away from land and again the general knowledge base disagreed with this. Only the commercial users agreed with this, theoretically, correct statement.

This perception or sentiment indicates that all users have a concept of the unknown related to climate change and the future, in general. Interestingly, when it comes to the uncertainties of the future, all users and subgroups agree that scientific reputation is important. This indicates that users understand that scientific rigour is needed to analyse

and accurately account for possible change. This is supported by the topical area postulations regarding institutional
reputation, scientific support and training. 100% of South African users, across both communities, agree that
training will be required in the future to help users understand the science behind ocean forecasts.

Although everyday use of the coastal ocean in South Africa is evident (de Vos and Rautenbach, 2019) the vast
majority of the public is not as closely linked with the ocean as Kiwis (New Zealanders) are (refer to Section 2).
This cultural difference was also observed in the present study where a greater contingency of the survey
participants in South Africa were commercial users. These also include members of the public who have a more
direct technical relationship with the ocean. Even though the New Zealand population is approximately 10 times
smaller than South Africa, the present study survey obtained approximately four times more interest in New
Zealand, illustrating the influential role of the ocean among New Zealanders. The distinct consensus patterns
obtained in this study present an image of South African users who are quite homogenous in their understanding
of salient forecast products and user needs. The New Zealand recreation cohort, though a remarkably heterogeneous
sector that includes a diversity of ocean uses, still exhibited a moderate-level agreement with the consensus model
(both in the country- and sector-specific models). It is noteworthy that New Zealand commercial users had weak
levels of agreement in all consensus models. This could be due to the larger range of participants (and thus ocean
activities), representing a wider variety of commercial users (refer to Figure 2, Question G).

One limitation of the present study pertains to the method with which the concepts used, as propositions in the
survey, were adopted. We used an expert workshop and literature review to brainstorm statements to include in the
survey. Although these statements were compiled based on previous first-hand engagements with users, and the
experts involved had many years of combined experience around the topic, the most ideal setting would have
involved dedicated focus group discussions or in-depth interviews with users to elicit a list of concepts for the
survey. Such a workshop was planned but made impossible due to the evolving covid-19 situation. The survey
represented what amounted to current thought on the subject, and these new perspectives from two southern
hemisphere countries, with different cultures, still demonstrated numerous coherent opinions and perceptions. The
valuable insights presented here are useful for both local and global forecast agencies who must cater for a global
market and public good.

### 5.1 A general conceptual user decision quality framework

To summarise the lessons learnt from engagement with the user of metocean information, the following conceptual matrix is presented. Here, it is asserted that users' decision quality is a function of the service provider's awareness of user needs and the accuracy, consistency, and salience (how forecast is packaged and communicated) of a product. Decision quality is defined as the users' ability to make informed decisions, correctly. Thus, the user is empowered to make the correct decision. This framework holds true for varying contexts of local and sectoral knowledge and general ocean literacy. In Figure 6 this conceptual framework is depicted schematically.

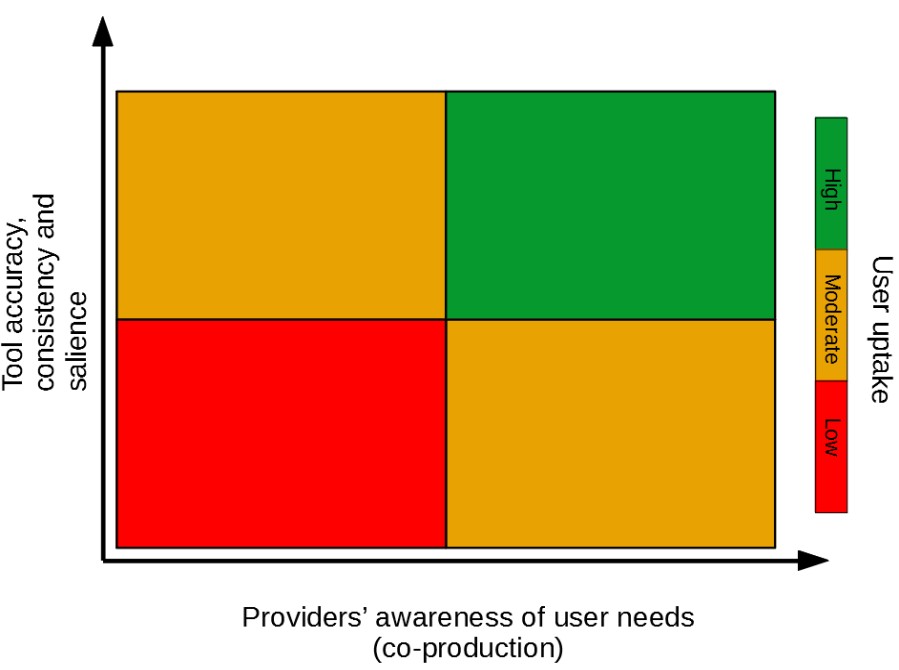

**Figure 6:** A conceptual matrix illustrating user uptake as a function of co-production and tool accuracy and consistency. This framework illustrates the co-dependence between science communication and bespoke, user centric, forecasting tools and products.

This conceptual model demonstrates the need for product and service co-production with users. While we established several important factors in forecast salience that can be classified under a global (cross-geographic, cross-sectoral) typology, other user needs were context-specific and/or were generated by varying degrees of ocean literacy. Service providers benefit from co-production as it can help to ensure that products are useful, usable and used (Vaughan et al., 2018). According to the respondents in the present study, considering rapid biophysical shifts

that are anticipated due to climate change, there is an increased need for science-based forecasts, and for greater
understanding of (and training in) forecasts and the science behind forecast services. This means that users can
benefit from collaboration with service providers through mutual learning, and the development of more bespoke
products. Investment in co-production can increase user trust in providers by increasing the transparency and
comprehensibility of forecast skill and relevant metrics. Our conceptual model can be applied to various locales,
industries, or interest groups, in deciding where the focus in new product development should be. For example, it
might be that whilst a product performs relatively well (high quantified skill level), local knowledge is lacking, and
this is the reason for poor decision making. As such, resources might be better spent addressing the local or sectoral
knowledge gap and ensuring that the product is used correctly, with appropriate regard for its limitations (Alexander
et al., 2020).

### 628   6. Conclusion

We used a consensus model approach to document and explore a potential typology of the factors that make
forecasts salient for users, in two southern hemisphere nations. In addition to these geographic settings, we also
explored the consensus around current and anticipated future user requirements in their sector-specific contexts.
Cultural consensus analysis allowed us to systematically explore regularities and variation in perceptions. We found
varying degrees of consensus among the whole group versus different subgroups of users. South African
respondents were homogenous in their agreement independent of sectoral affiliation. New Zealand's recreational
users were in moderate agreement amongst themselves and with South African user groups, but commercial users
were divided. For all user groups, ease of use, customizable features, consistency and accuracy were some of the
important factors in service uptake, however established reputation of the provider was important specifically in
the commercial users and South African respondent cohorts. Respondents emphasized a number of priorities for
science-based forecasts in the future (in light of anticipated climate change impacts). Based on our findings we
proposed a decision-quality framework schematic that 1) builds on the global dimensions of established user
requirements and 2) emphasizes the role of co-production in generating context-specific knowledge. We aim to
bring prominence to the need to move to demand-driven models of service development by reworking the user-
provider relationship. Going forward, future work could extend the consensus method toward evaluating the risks
and uncertainties that are priority to different user groups, and which services are most relevant and/or lacking to
reduce those uncertainties. Co-production may help to operationalize such practical evaluations of risks, and of the
evaluative criteria needed for a comparison across multiple settings and contexts for better service provision. While
co-production may not always be the desired approach (especially when the problem uncertainty and/or service
demand are low and supply-driven solutions suffice). But when many users are impacted, and uncertainties are
high, user collaboration helps to ensure product salience, the eventual uptake of services, as well it adds value to
the forecast value chain by supporting and promoting safe marine activities.

**Appendix 1**

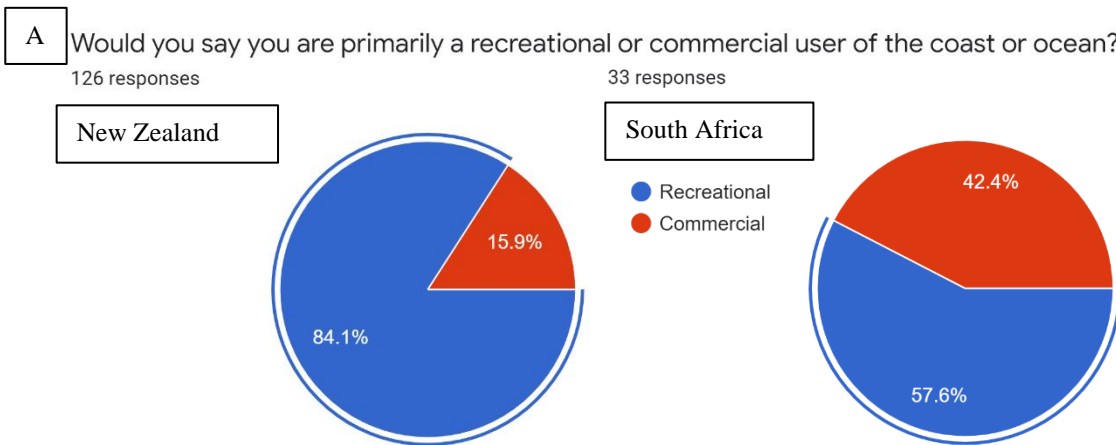


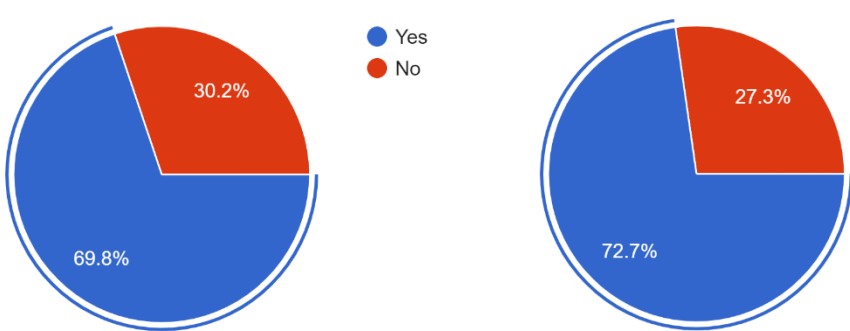


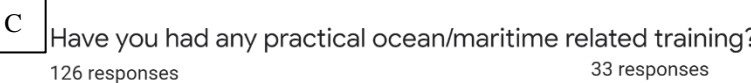

C Have you had any practical ocean/maritime related training?
126 responses                                    33 responses

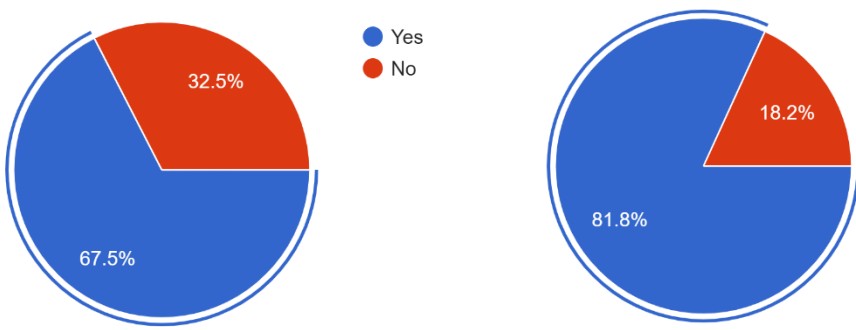

655

**Appendix 2**

Noteworthy variations in Table 1 present the following patterns in the various consensus models:

Whole group consensus model: The 31 participants from South Africa (Mean = 0.61, SD = 0.12) compared to the 126 participants from New Zealand (Mean = 0.51, SD = 0.18) demonstrated significantly higher average competence score, t(155) = 2.8, p = 0.0056. There was no significant effect for sectoral affiliation. Out of 157 respondents, one had a negative competence score close to zero (-0.063). While these results suggest an overall shared knowledge domain regarding user needs, the significant variation in mean competence scores between the two countries means there are some issue areas that split perspectives between country-specific user contexts.South African consensus model:  There were no negative competence scores, and both commercial and recreational user subgroups attained similar mean scores (~ 0.6.) This subgroup's consensus model shows high levels of agreement among respondents, and the agreement bridges across commercial and recreational users. New Zealand consensus model: three respondents had negative competence scores. Two of these were close to zero (-0.053 and -0.003) and the third ~0.1. The overall mean consensus score was moderate at 0.5, and the difference between commercial versus recreational user average scores was not statistically significant. However, the commercial group's 0.4 average indicates low levels of agreement in this subgroup with a potential consensus model. It is difficult to definitively infer the existence of a clearly defined cultural pattern in this case: some of the assumptions of a cultural model are met (eigenvalue ratio > 3.0) but three negative competence scores (even if two are very close to zero) speak to a contested consensus domain, though large parts of the mental models may overlap between subgroups.

Commercial users' consensus model: the mean group competence score was moderate at 0.52, with one respondent attaining a negative competence score near the ~0.1 level. The 14 respondents from South Africa (Mean = 0.62, SD = 0.12) compared to the 20 respondents from New Zealand (Mean = 0.44, SD = 0.23) demonstrated significantly higher average competence scores, $t(32) = 2.572$, $p = 0.015$. Seven of the twenty participants from the New Zealand subgroup had a competence score below 0.4 (including the respondent with the negative score), and a moderate but notable variability (SD +- 0.23) in individual competence scores. Despite the sufficient eigenvalue ratio, the significant variation in mean group scores between commercial users from the two nations and low competence scores in one subgroup suggest a noncoherent consensus model in this sector that does not span well across the geographic divide. Recreational users' consensus model: the 17 respondents from South Africa (Mean = 0.62, SD = 0.13) compared to the 106 respondents from New Zealand (Mean = 0.52, SD = 0.17) demonstrated significantly higher average competence score, $t(121) = 2.193$, $p = 0.03$. Despite these variations the New Zealand cohort's mean score shows moderate agreement with the consensus model, and there was only one, near-zero negative competence score (-0.009). In this sector, there is a moderate level of agreement in the consensus model between users in the two countries.

## Author contribution

Dr. Rautenbach conceptualized the original idea. Dr. Blair helped in evolving the original idea into the resulting study and co-developed the aims and goals of the study. Dr. Rautenbach took the lead in data curation. This entailed distributing the survey, collecting the responses and summarising them into an initial digital format. Dr Blair took the lead in the formal analysis, doing the CCA analysis and producing the quantified results. Dr. Rautenbach secured the funding for the study. The investigations of the study were done by both Dr. Rautenbach and Dr. Blair through numerous online discussions. The methodology was jointly developed but with Dr. Blair taking the lead with the CCA analysis. Dr. Rautenbach took the lead in project administration and providing sector specific insights. Both authors contributed to data visualisations. Both authors wrote the manuscript. Dr. Rautenbach wrote the original draft with significant contributions from Dr. Blair during the reviewing process.

## Acknowledgements

The authors would like to thank Marc de Vos, senior scientist at the South African Weather Services, for distributing the survey in South Africa. Mr. de Vos also conceptualised the first version of the general conceptual user decision quality framework and helped in formulating some of the research question propositions. We also

acknowledge the MetService of New Zealand who helped distribute the survey in New Zealand, while Dr.
Rautenbach was still an employee.

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
