# Peer review of "Marine Meteorological forecasts for Coastal Ocean Users - Perceptions, Usability and"

_Geoscience Communication, 2020_

## Author Comment (AC1)

Please find our inline response following each comment.

Overall, I like the manuscript. For example, I think it is a good idea to use an earlier study as a starting point, and compare with those results, as well as a comparison between two different countries on two different continents. I also like connections drawn between the perceptions and cultural similarities/differences, and I think some of the discussions related to these issues are new and interesting. Further, I like the idea of having a workshop with metservices to discuss topics and questions when preparing the study/survey. I do think that such a paper fits within the scope of the journal and will be interesting to readers of Geoscience communication as well as to the wider meteorological society. However, I also have some issues that I think the authors need to sort out before the manuscript is ready for publication (major revision required). Below you will find my comments. They are numerous, but you should not be scared; I'm not negative to the manuscript, I'm just engaged in the topic and want to contribute such that it can make a higher impact.

Authors' response:

Thank you very much for your kind words. This has truly been a wonderful interdisciplinary study which we hope will have positive ramifications for the uptake of metocean forecasts.

General comments

1. To me, and others unfamiliar with the methodology used in the study (I assume many from the meteorological services are), it would be good to get a better idea of the concept/method earlier in the manuscript. There are many paragraphs and sections mentioning "CCA" and "(shared) knowledge", but I don't really understand what it is before late in the results section when more concrete examples are given. Perhaps the authors could consider including an example of the result and which knowledge they are talking about already in the abstract. Now the abstract is describing the methodology, but little is said about the results or impact of the study, so including an example of a result could strengthen the abstract anyway, I think. The same can be said about the results section, which is interesting, but is a bit technical or abstract to people not familiar with CCA. Actually, it is not really before I read section 4.2.2 I started to understand what this actually was about. Then it also connected better to the research questions, than earlier in the results section. Again, it  would help the reader better understand if there were some concrete examples of knowledge types/impact factors/questions etc. early on and/or throughout the manuscript, to make it easier to read and understand.

   Authors' response:

   Thank you for these helpful comments. In order to give clarity to the method and process, we reorganized the Introduction and Background sections to bring forward the Aims section, where we also added some research questions that

align with the conceptual framing and the method (cultural consensus analysis). This is hopefully helpful in linking objectives and questions with the method.

Furthermore, we added into what is now section 2.2 the following blocks of text with new references (highlighted), these are also visible through the track changes document.

*"The type of relationship users cultivate with the ocean, and the resulting information need that is generated, is not only driven by geographical contexts but also by sectoral differences that determine sociomaterial (linked human-technological) settings (Blair et al., 2020; Lamers et al., 2018). Marine meteorological forecast users engage with metocean information as a tool to mitigate risks. Attitudes toward risks are a result of a constellation of individual and cultural factors, tied to bias, attitudes, preferences as well as societal influences and dominant worldviews (Fischhoff et al. 1978; Douglas and Wildavsky 1982; Lichtenstein and Slovic 2006; Kahan et al. 2012). These attitudes together can have a profound impact on the type of weather and climate information sought for decision making (O'Connor et al.2005; Kirchhoff et al. 2013). We also know that mariners and the organizations underlying navigation develop distinctive traits based on unique mental models, organizations and decision-cultures (Lemire 2015; Kuonen et al., 2019; Hederstrom n.d.) and these factors uniquely impact mariners' information needs (e.g. Wagner et al, 2020). Forecast services are used in distinct ways in different sociomaterial settings, and these differences impact the temporal and spatial scale at which information is needed for planning and for tactical decisions. Consequently, the socio-economic value that may be derived from salient forecasting services varies across a wide spectrum of geographic and sectoral contexts as well."*

and:

*"As more interdisciplinary research includes diverse stakeholders and their observations about the technical, natural and human factors that drive the need for information, it is increasingly apparent that understanding user needs, often in cross-sectoral and cross-cultural settings, is a significant challenge. Culture affects users' perceptions about, and attitudes toward, technologies in general (Lee et al. 2007; Lim & Park 2013), and the meaning and relative importance of salient scientific information (e.g. Martinson & Westwood 1997).Traditional interview and questionnaire methods do not always explain the variation in experiential knowledge that may exist across representatives of a wide range of sectors and decision environments. We used Cultural Consensus Analysis (CCA) (Romney et al., 1986) to document this variation and to look for patterns in user perceptions about the important factors that make forecast products trusted and used."*

and into Methods section, 1st paragraph:

*"An advantage of cultural consensus analysis is that a small population of respondents can yield rich observations and data about sector (commercial and*

*recreational) or locality-specific (South African and New Zealand) views and knowledge domains as they may exist among participants."*

and in Section 4.2.1, 2nd paragraph we modified the sentence for clarity about group-level knowledge and the consensus model:

*"A group's average estimated knowledge score above 0.5 indicates moderate agreement in the group, pointing to about an underlying model of shared knowledge (Weller, 2007). An individual's competence (or knowledge score) is the probability that the informant knows (not guesses) the answer to a question, and it is a value between 0 and 1."*

These steps should help to solidify the links between our aims and the methodology used, as well to familiarize the reader with the language.

2. Related to this is another issue, I think the manuscript would become stronger if there is a better connection to the research question throughout the whole manuscript. My impression through the title, the aim section, and the research questions in 3.1, is that the study is mainly about forecast perception and factors driving uptake of information. This fits quite well with the topics discussed in the introduction. However, section 2, which I find interesting, is more about cultural dimensions, naturally the same with the methodology section (CCA) and discussion.

Authors' response:

The cultural dimensions establish the "knowledge groups" or the cohesive, group-culture thinking about the issues at hand. Once we establish that these shared knowledge groups exist (or not) based on locality and sectoral affiliation (section 4.2.1) then we present what exactly each group thinks about the issues (section 4.2.2). We made an effort to better organize these thoughts and added the following text at the end of Section 4.2.1 as a transition to Section 4.2.2:

*"In the next section we present the answers (the consensus results) in each group of analysis, for a comparative analysis of the ways in which locality (national affiliation) and sectoral affiliation resulted in the same or different answers to our questions."*

Also, parts of the results (especially 4.2.1) are a bit technical and focus differences between groups and demographics, whereas the research questions are much more concrete about perceptions and usage - section 4.2.2 is more like what I expected to read. There might be something I miss out, or need to read more carefully, but at the moment I am wondering if the authors could consider to better connect what I believed the study was about (perceptions and uptake) and what I think part of the study is focusing (culture and differences between groups) such that it become more coherent. Maybe it needs some

restructuring, or maybe it just needs to make it clearer through the title, introduction of research questions.

Authors' response:

As we responded above, both aspects (culture and perception/usage issues) are linked and important: we first establish the existence of knowledge/perceptions that are driven by geographically or sectorally determined "cultures" and then in Section 4.2.2 show how each "culture" responded similarly or differently to the same questions. To this end, we completely reorganized and cleaned up Section 4.2.1 to shorten it and make it less technical, with a transition sentence to link the culture section to the survey results section in Section 4.2.2.

In order to ease the reading of Section 4.2.1 by making it less technical, and to make it more concise - focusing on only the most relevant information to support conclusions in the main text, we moved a large block of explanatory text into what is now Appendix A. This block of text explained in great detail the patterns of variation in Table 1, but the Figures (2-3) that follow and the analysis of those scatter plots serve the same purpose. This made those several paragraphs superfluous, though someone specifically interested in the methodology may wish to overview this info as an Appendix.

Furthermore, we added this explanatory text in Section 4.2.1 paragraph 4:

*"Those who had high levels of agreement with each other are situated close to each other, while those who had high levels of disagreement are scattered proportionally farther apart."*

to assist the reading/interpretation of the scatter plots.

3. There are recommendations about how to provide information, but they are quite general (easy navigation, few clicks, accurate forecasts etc.). Hence, I am not sure how useful they are to service providers developing platforms. How many clicks are few clicks? What is easy navigation, and for who? What is an (in)accurate forecast? This makes me wonder who the study is for, operational people or researchers? If the study is aimed at operational meteorological services, to improve value of metocean forecast information, perhaps the study would have a bigger impact if more concrete recommendations can be suggested (if possible, given the data the authors have), and (as explained above), more examples related to the CCA is made throughout the manuscript. I am also wondering if it is possible from Table 2 to see which one is more important to the participants (is it related to the percentages)? e.g. visual experience and number of clicks can be leading the service provider in the same direction, but it can also be a contradiction or dilemma, and which one should they then focus?

Authors' response:

It is difficult to also go into detail with regards to how many clicks are too much. That will basically be an entirely different study and will also differ between user groups. The aim was rather to make it clear to service providers that these aspects of platform development are important and must be part of the initial design as well the construction of tools and platforms. That is where the usefulness lies. It will not be possible to say as an example: "3 clicks are the maximum", as that is a whole new layer of information and would have made the survey far too cumbersome. Thus, depending on the particular tool these parameters must then be minimised, also taking into account the target audience.

With regards to inaccurate forecasts - this is also not linear or trivial to define. Numerical model skill and accuracy definitions are a whole field by itself. They also differ between the atmosphere and the ocean and the particular parameter. In the present study we were rather interested in how important accuracy is for users (in whichever way they define it). If we went into too much detail in this regard the survey would again become too cumbersome and we would have struggled even more to get enough participants. A question regarding accuracy is important for all users, as even recreational users do have their own concept of accuracy (or reliability).

To ensure that the data of the present study is better understood, a more in detail explanation of Table 2 was added (where all the data have been summarized).

Table 2 caption has been updated to better explain the table for the reader, and Section 4.2.2 starts with the following amended text:

*"Table 2 presents the results of the survey. These are the direct questions and resulting propositions that were distributed in the survey and form the basis of the present study. The column titled "whole-group CCA" is based on the consensus analysis of all respondents together, and it shows the aggregate group belief (culturally-correct answer) with either agreement (green icon) or disagreement (red icon) with the propositions. The other columns indicate the percent frequency of matching answers (or agreement with the whole-group CCA), in each subgroup. In case a subgroup's own consensus-model (consensus analysis run only including its members) produced a group belief that deviates from the whole-group CCA, the added icon indicates the correct answer in the sub-group."*

Specific comments

1. This relates to the survey or the questions. When I read, some data/results come a bit surprising, because I couldn't see the questions or topics being mentioned explicitly earlier. For example, in line 442, it says that the participants were questioned about their trust of their own NMHSs. Where can I see that question?

Is it part of table 2.2 (are all questions asked there), or are additional questions also asked but not shown? That should be made clear in the manuscript.

Authors' response:

Some questions did not fulfil the criteria for CCA. Thus, their results were not added to the manuscript. The raw survey results we deemed still valuable and thus chose to convey this information in the text. To ensure this is clear to the reader a paragraph has been added at the end of Section 4.1, explaining the caveats of the additional data presented there.

2. I am a bit surprised that uncertainties related to climate change are part of the research questions. The title and introduction does not really give an impression that climate change is a topic, and I find little about that in the results as well, I think (until the end of section 4.2.2). Still, it is part of the discussion and conclusion. I think it needs to be clearer in the introduction that this topic is part of the study, since it does not directly relate to the perceptions and uptake of forecast information, at least not to me.

Authors' response:

The climate change topic has now been added to the abstract:

*"We discuss the implications from our findings on important factors in service uptake, and therefore on the production of salient forecasts. Several priorities for science-based forecasts in the future are also reflected on considering anticipated climate change impacts."*

In the background of the study:

*"With the continuing rise in Climate Change impacts and changing weather patterns, user understanding, and uptake of forecast products have never been more important (a sentiment echoed in the results of the present study). Here, we will focus on ocean and coastal users and include marine forecasts as the main predictand."*

In Section 3.1 the fourth point also highlights uncertainties regarding future forecasts. Here some text has been added to ensure the reader understands we are referring to Climate Change (CC). It is also important to remember that users' perceptions of CC will play an important role in estimating the relevance of forecasting (and science) in the future. Through this connection CC is thus related to perception and uptake as the environment is already changing.

3. Line 98-99: A distinction between specialist users and the public is made, and I agree level of understanding can be an explainer. However, maybe it is worth mentioning that it can also be easier to agree upon communication with a specialist user group than the general public? Another point (related to this and lines 230-235, which I agree upon) is that some commercial users want/need to be efficient, they are not interested, it is just part of their work, whereas some recreational users are really interested, they don't need to be effective. Hence, a

strict categorisation is difficult, in some situations a person can be an interested specialist, in others spend little time. It can depend on the task, not only the person or group. (see the first paper suggested below for more details.)

Authors' response:

The following sentence has been added to Section 2.1:

*"Doksæter Sivle and Kolstø, (2016) investigated the use of online weather information for everyday decision making. Here it has become clear that this distinction is also dependent on the task (for which the forecast is used) and not only on the person or group."*

4. Line 143 - would be nice if the authors could consider to give an example of language weaved into ocean-based references and symbolism.

Authors' response:

A sentence has been added explain the Mangopare, which also features in the MetService's logo.

One such example is the Mangopare (hammerhead shark symbol). The double Mangopare has been incorporated into the New Zealand MetService's logo and represents weather prediction and oceanography and their dependence on each other.

[Figure]

5. Line 147 - explain short who Khoisan people are?

Authors' response:

The following sentence and reference have been added:

*"Here Khoisan refers to the first indigenous peoples of Southern Africa (Rito et al., 2013)."*

6. Research question 3.1: Studying forecasts (the content/information) or forecasting platforms? Could be worth clarifying throughout the manuscript.

Authors' response:

This has been clarified

7. I am sure it is a sufficient number of participants according to method, but still 31 from South Africa is not much in terms of absolute numbers. I just want to ask the authors to be careful in their language so they do not generalize the findings, e.g. line 257 ("while South African users" - should be participants?).

Authors' response:

Line 257's "users" have been changed to "participants".

8. Figure 1G and line 280: It might be me not reading the text well enough, but is it clear whether these groups include both commercial and recreational users, or one of them (e.g. windsurfers or people having a windsurfers rental; commercial fishers, or people having fishing as a hobby etc.)?

These were generalised questions that all participants answered and thus these are all the participants. Clarifying text was added on the original line 357.

9. Lines 285-290: If possible, a map showing the areas would be useful.

Authors' response:

The new Figure 2 has been added.

10. Line 465: A good explanation of what a red cross in Table 2 actually means. Perhaps something similar can be given earlier in the manuscript, to make it easier to read Table 2?

Authors' response:

We added the following block of text (highlighted) in the 1st paragraph of Section 4.2.2:

*"The subgroups' consensus models can be found under appropriately titled columns. Where the subgroup's own consensus-model matched the whole-group CCA, we provide the Values are percent of responses that matched the whole-group CCA to show level of agreement. Where a subgroup's own consensus-model deviates from the whole-group CCA, the added icon indicates the correct answer in the sub-group. "*

This, along with Table 2's caption, will hopefully make it clearer to readers how to interpret the symbols and numeric values in the table.

11. Line 512: There are cultural differences, yes. I am speculating - perhaps you know - are there also other differences (economical) steering if people have recreational use of the coast?

Authors' response:

Very good points and might also be true. This might add another layer of detail to the results that we did not tease out currently and falls a bit outside of the scope of the present study.

12. I want to **suggest a few references that** might be of interest to the authors:

1. Doksæter Sivle, A., & Kolstø, S. D. (2016). Use of online weather information in everyday decisionâmaking by laypeople and implications for communication of weather information. Meteorological Applications, 23(4), 650-662.

2. In Norwegian, a user survey from 2015 related to marine services (https://www.met.no/publikasjoner/met-info/met-info-2015, number 15/2015, Gjesdal et al.)

---

## Author Comment (AC2)

Thank you for taking the time to review the manuscript.

Thank you for the opportunity to review this manuscript.

You have produced a paper on a very pertinent topic that, as you correctly identified, has not received a lot of attention thus far in the academic literature but is a topic that is highly relevant among specific communities of practice - in your case, the marine user community. I enjoyed reading about the approach you have taken and the potentially very promising comparative analysis.

Thank you for the kind words.

However, I was somewhat disappointed by how sloppily the manuscript was put together. There are numerous typos, punctuation errors and mistakes in syntax and grammar (see e.g., ll. 9, 20, 26, 71, 97, 108, 126,172, 256, 342, 345, 356, 357, 371, 403, 439, 463, 465, 471, 473, 477, 540 or 546), which thorough proof-reading should have revealed, especially since the lead author seems to be a native English speaker originally from South Africa. The abstract seems to have been put together in a hurry and is rather disjunct and unconvincing, especially since relevant context is not provided (e.g., what is the "proposed disconnect" you are referring to, and who "proposes" its existence?). An abstract should contain a short synopsis of the key conclusions reached, which are currently not highlighted in the abstract. Sadly, you are not selling your research convincingly if you do not place enough emphasis on a coherent, concise and convincing abstract.

Authors' response:

The manuscript has been proofread again. All the specific language alterations have been made. It should also be mentioned that the lead author is indeed not a native English speaker (nor the second author). In fact, less than 10% of South Africa are native English speakers.

The abstract has been restructured based on Reviewer 1's comments as well. We have now also incorporated these comments into our abstract and conclusion with clear highlights of the study conclusions.

The contextual section 2 of the MS would be better placed in the introduction and before the aims to allow for a better flow from the aims to the methods section. Also note that what you present is a methods section rather than a methodology per se. Section 2 and the Introduction appear to simplify a bit too dramatically, at least for my taste, the character and history of New Zealand and South Africa. Claiming that the two countries are climatologically similar and at similar latitudes is like stating that France and Algeria are at similar latitudes. There are significant latitudinal differences between S.A. and NZ, with the latter also consisting of more than 2 islands as you claimed on p. 5. It might also be worth noting that one is an island state (NZ) and the other is not (S.A.) You refer to the coastlines of these two countries as "extreme", which made me wonder what you meant by that term.

Authors' response:

The manuscript has been restructured to try and take into account both reviewers' comments. The methodology section has been renamed methods and the introduction has been restructured. Clear reference is made that New Zealand is an island state. It's also clearly stated that the two countries don't have the same climatologies but due to their location (exposed to the Southern Ocean) there are indeed very strong similarities. Hence, it's not really the same as saying France and Algeria are the same. Having worked in both countries' MetServices I can indeed confirm this to be true. With regards to the coastlines: New Zealand has the 17th longest coastline in the world while South Africa is at number 49. Thus compared to most countries in the world these two southern hemisphere countries have long coastlines. The word "extreme" has been replaced with "extensive".

We were not aiming to provide a comprehensive comparison between the two countries' climatologies. That will far exceed the page limit and distract from the focus of the study. We rather aimed to introduce the reader to some of the similarities between the two countries (that we as South Africans and Kiwi experience). The two countries have strong social links as South Africans now make 1.36% of New Zealand's population. From this viewpoint we hoped to help the reader get a shortcut into our viewpoint and insights.

The methods section raised more questions than it answered. You are posing some interesting and certainly relevant research questions but

it is not clear whether these were the questions guiding your overall work and the design of the questions, or whether they were the questions asked in the questionnaire itself. The latter seemed to be the case, which I have not seen done in other survey questionnaires. [...] The research process you describe in your methods section left me somewhat puzzled - why was your initial step to formulate research questions, when these should have been identified well before you even set out to think about the survey as surely the questions you ask would determine the most appropriate methods to employ rather than the other way around? [...] You should also expain how many questions you asked in the survey, what type of questions,

Authors' response:

Our main research questions (Section 1.1 Aims), specifically Q1 and Q2 resulted in sub questions, which we used to construct the survey instrument.

The constructs / concepts that our propositions asked about, were workshopped with metocean researchers. We did so to make sure we are asking the right questions, that we are including questions about a diverse set of factors that may be important to users.

Actions taken:

We revised Section 3.1 to clearly describe the process of how the survey was designed, how overarching survey questions relate to main research questions, how the propositions were designed, their number and format.

You also claim to have aimed to "test the differences between the social norms, values and attittudes" (l 170) in NZ and S.A. user groups. This would have been very interesting as well as ambitious, but nothing of what you present in the MS links back to norms, values and attitudes. In fact, you do not provide a definition of these terms nor do you apply any of the established psychological tests that are typically used to explore these aspects.

Authors' response:

In fact in Section 2.2 (and to some extent in 2.1 as well) we establish via an in-depth discussion the role of mariners' operational contexts as culture in the development of group-specific meanings, norms and understandings in our target population, and that these have bearing on risk attitudes and risk perception (also that information has a role in mitigating such uncertainties). Our focus is on finding group-level meanings, as opposed to individual ones. We state:

"*Attitudes toward risks are a result of a constellation of individual and cultural factors, tied to bias, attitudes, preferences as well as societal influences and dominant worldviews (Fischhoff et al. 1978; Douglas and Wildavsky 1982; Lichtenstein and Slovic 2006; Kahan et al. 2012). These attitudes together can have a profound impact on the type of weather and climate information sought for decision making (O'Connor et al.2005; Kirchhoff et al. 2013).*"

For this reason, we do not engage in psychometric testing specifically because our aim wasn't to reveal the suite of psychological factors that shape individual coastal users' risk perceptions, or ways in which forecasts can reduce perceptions of risks for individual users who share certain psychological traits. As we put forth (Section 1 Introduction, last sentence):

"*The effectiveness with which relevant information is communicated to those clients can differ depending on the user's domain knowledge and the utilisation purpose (e.g. Kirchhoff et al. 2013; Lamers et al. 2018; O'Connor et al. 2005; Wagner et al. 2020). Specific clients often require bespoke solutions not entirely transferable to other users.*"

In other words, our research instead focuses on whether specific groups of users' domain knowledge stemming from different operational contexts (commercial/recreational and South Africa/New Zealand) indeed result in unique group-level perceptions about forecast usability. We aim to produce actionable results for forecast producers by informing them ways in which membership in (cultural. geographical) groups impact forecast salience. We do not explore individual personality traits.

Actions taken:

To highlight the connection between the selected method and the theoretical discussion in Section 2.2, we added the following under Methods:

"*Mariners' unique mental models, organizations and cultural domains result from specific practices and operational contexts (Section 2.2). Cultural consensus is an appropriate method to assess cultural domains (Romney et al. (1986)); in this case gauging the extent to which the practices and ocean use contexts of recreational marine users are of the same cultural domain -i.e. they develop and share the same understandings about the factors that enhance forecast usability- as professional mariners.*"

We altered the sentence Reviewer referred to "test the differences between the social norms, values and attitudes" (l 170)" the sentence refers instead to "knowledge domains" to more accurately refer to our theoretical and methodological commitments.

We added an explanation to our use of culture in the context of our consensus analysis, section 2.2:

"*In this research we use the term culture to denote learned ways of knowing; more specifically, learned knowledge that shapes people's approach to ocean resources, and ocean information use.*"

how you distributed the survey and to whom

Authors' response:

The survey was advertised through mailing lists. These included numerous user specific community mailing lists in both South Africa and New Zealand. Both the MetService and SAWS also advertised the survey on their websites. We also personally distributed the survey through our social and professional networks. This was also described in the original manuscript.

and most importantly, how you obtained informed consent and protected the rights of the participants. You should also note where you obtained Human Ethics Consent for the survey. With the lead author based in New Zealand, I take it as a given that this has been done and it should be noted in the MS, even if only in form of a footnote.

Authors' response:

Institutional Review Board approval is mandated either at the institutional level (neither author's institution requires a formal ethical review) or by funders (this also did not apply in our case). For this reason, there was no ethics review conducted for this survey.

The authors are well-familiar with ethical conduct however (e.g. the second author worked for years with Alaska Inupiat communities and went through numerous IRB approvals in the process) and our subjects were well-protected. The surveys were anonymized, and no personally identifiable information was collected from participants.

As the data analysis is concerned, the readers would appreciate learning a bit more about Romney et al.'s (1986) consensus model and whether it is still considered cutting edge. What is a UCINET software package?  What were the limitations of your analysis and research approach?

Authors' response:

We added in the Methods section the below listed collection of sample literature to demonstrate the wide-ranging applicability of cultural consensus analysis. We trust that it is sufficient to demonstrate the usefulness of the method. It is cutting-edge to the extent it is still used successfully to produce valuable research output, and the underlying analysis (principal component analysis) isn't expected to fall out of favour in the academic community. The software used was mentioned only to the extent it is a standard practice in literature where software is used, for better transparency, and replicability.

As for limitations, we added the following text in the Discussion section, last paragraph section 5:

"*One limitation of our study pertains to the method with which the concepts used as propositions in the survey were adopted. We used an online expert workshop and literature review to brainstorm statements to include in the survey. Although these statements were compiled based on previous first-hand engagements with users, and the experts involved had many years of combined experience around the topic, the most ideal setting would have involved dedicated focus group discussions or in-depth interviews with users to elicit a list of concepts for the survey. Such a workshop was planned but made impossible due to the evolving covid-19 situation. The survey represented what amounted to current thought on the subject, and these new perspectives from two southern hemisphere countries, with different cultures, still demonstrated numerous coherent opinions and perceptions. The valuable insights presented here are useful for both local and global forecast agencies who must cater for a global market and public good.*"

Sample CCA literature cited in Methods:

Natural resource management:

Miller M.L., Kanko J., Bartram P., Marks J. and D. Brewer. 2004. Cultural consensus analysis and environmental anthropology: Yellowfin tuna fishery management in Hawaii. CrossCultural Research 38(3):289–314.

Naves, L. C., Simeone, W. E., Lowe, M. E., Valentine, E. M., Stickwan, G., & Brady, J. (2015). Cultural consensus on salmon fisheries and ecology in the Copper River, Alaska. Arctic, 210-222.

Traditional / local / lay / expert knowledge

Medin, D. L., Ross, N., Atran, S., Burnett, R. C., & Blok, S. V. (2002). Categorization and reasoning in relation to culture and expertise. In Psychology of Learning and Motivation (Vol. 41) (pp. 1–41). Cambridge, MA: Academic Press.

Reyes-García, V., Huanca, T., Vadez, V., Leonard, W., & Wilkie, D. (2006). Cultural, practical, and economic value of wild plants: A quantitative study in the Bolivian Amazon. Economic Botany, 60(1), 62–74.

Van Holt, T., Bernard, H. R., Weller, S., Townsend, W., & Cronkleton, P. (2016). Influence of the expert effect on cultural models. Human Dimensions of Wildlife, 21(2), 169–179.

Public health:

Garro, L. C. (1996). Intracultural variation in causal accounts of diabetes: a comparison of three Canadian Anishinaabe (Ojibway) communities. Culture, Medicine and Psychiatry, 20(4), 381-420.

Weller S.C. and Baer R.D. 2001. Intra- and intercultural variation in the definition of five illnesses: AIDS, diabetes, the common cold, empacho, and mal de ojo. Cross-Cultural Research 35(2):201–226

Weller, S. C., Baer, R. D., de Alba Garcia, J. G., & Salcedo Rocha, A. L. (2012). Explanatory models of diabetes in the U.S. and Mexico: The patient–provider gap and cultural competence. Social Science & Medicine, 75(6), 1088–1096. https://doi.org/10.1016/j.socscimed.2012.05.00

Strong, A. E., & White, T. L. (2020). Using paired cultural modelling and cultural consensus analysis to maximize programme suitability in local contexts. Health policy and planning, 35(1), 115-121.

Tourism:

Paris, C. M., Musa, G., & Thirumoorthi, T. (2015). A comparison between Asian and Australasia backpackers using cultural consensus analysis. Current Issues in Tourism, 18(2), 175-195.

Ribeiro, N. F. (2016). Do tourists do what they say they do? An application of the cultural consensus and cultural consonance models to tourism research.

The figures you include in the paper should ideally be formatted in such a way that the different colours used in the pie charts can be easily distinguished. This is presently not the case, and I suggest linking the categories directly to the respective pie sections to make this clearer. You might also wish to separate the separate questions that you are presenting as subsections of Figure 1 into stand-alone figures, which would make referencing to specific parts of the figures much easier.

Authors' response:

Unfortunately, we are not sure how to change the figure as the figure colours are indeed directly linked with the categories. We envisaged the referencing to be of the format: Figure 1 A, for example, and do feel it is much better than having a list of figures and captions in series below each other. All the possible figure colours have already been used. Hence why we also list and name the major statistics in the text.

Nevertheless, the following figure was added to replace Figure 1G.

[Figure]

In l 255 you claim that "[f]rom these results it seems that most people will only look at a forecast once a day..." - nothing that you present in your MS shows that you can make such a conclusion, and you may wish to examine a bit better how you derive this conclusion. You argue that "[f]air representation was also received from the other districts" (in New Zealand) (l 287) but with 87% of your participant from Auckland, the Waikato, Wellington and Northland, this is not feasible.

Authors' response:

Here we would like to refer to Figure 1 D where most participants indicated that they only look at the forecast daily. These statistics are representative of the population distribution

as well. 33% of New Zealand lives in Auckland, so to have other districts presented in the survey we considered to be fair. I guess it's subjective. Nevertheless, the word "fair" was removed.

Considering the relatively low number of participants, especially from South Africa, it would be prudent to discuss whether the results are actually comparable. You seem to simply assume that they are, but you do not even seem to have statistically significant sample sizes from S.A., especially not when diving them into recreational and commercial users.

Authors' response:

We ended the methods section with an explanation of the circumstances in which the CCA method is robust, in spite of small sample sizes, or differences in sample sizes between groups. We write (Section 3.2):

"*The minimum sample size required for the consensus model depends on the level of agreement, the number of informants, and the validity of the aggregated responses (Weller, 2007). For example, at a low-level agreement of 50% (mean competence or knowledge score of .5) at .95 validity the minimum sample size is twenty-eight people per group. The same at 60% agreement is seventeen people.*"

And in Section 4:

"*Because of the level of agreement (mean competence scores) and eigen value ratios obtained in the New Zealand and South African cohorts, it was possible to establish consensus models despite the different participation rates (refer to Section 4.2).*"

*Weller (2007) establishes that when the level of shared beliefs reaches the 50% level (average knowledge score is .50), there is sufficient agreement that the consensus model can identify a single response pattern. The minimum sample size, given 0.5 knowledge score at .95 validity, is about thirty people (per group)*"

The precise number of required participants established in Weller (2007) for these criteria is n=28. All our groups which we subjected to separate testing, met these conditions as laid out in Table 1:

All participants evaluated together: N = 157, competence 0.53

South Africa: n = 31 competence 0.61

New Zealand n = 126 competence 0.5

Commercial users (SA and NZ): n = 34 competence 0.52

Recreational users (SA and NZ) n = 123 competence 0.53

The other values in Table 1 (knowledge scores for even smaller groups such as South African commercial users n = 14) are not part of separate consensus analyses. For these

small sub-subgroups, we did not calculate consensus. These values simply show the breakdown of mean competence scores. For example, all commercial users were evaluated for consensus analysis together (Table 1, Row 5). The mean competence score for this group is shown, as well as the SA and NZ cohort separately for comparison. But these are all a part of the same consensus cohort/analysis/model that was run on n=34 or all commercial users together.

Because the precise details of the method are complex, and the underlying functions are built into UCINET, the software used by everyone who performs CCA (for example the Spearman-Brown prophesy formula is used to articulate the mathematical relationship between the number of people, the agreement among those people, and the reliability (and validity) of an aggregation of their responses), we determined that an overview of the CCA basic steps in Methods, but without the ins-and-outs, is most useful for the readership.

Action taken:

We realize that this method is nuanced. We added some text to help the reader:

1. added the following text in section 4.2.1:

"*Further studies are needed that explore the knowledge domain of New Zealand commercial users, with regards to forecast needs and perceptions about existing services. In this study the number of participants in this cohort was too low for a separate consensus analysis. For now, we can conclude that this cohort's understanding on the issues did not conform well to that of other cohorts (Table 1).*"

2. Modified the beginning of results (new text in bold):

"*These numbers proved to be sufficient for the use of CCA because the level of agreement (mean competence scores >= 0.5) and eigen value ratios (> 3.0) obtained in all cohorts (whole-group, New Zealand, South Africa, commercial users, recreational users) meant that a consensus model can be established with just twenty-eight people per group (Table 1). It was possible to establish consensus models despite the different participation rates and small sample sizes because in CCA validity is also a function of level of agreement (Weller 2007).*"

3. Added the following clarifying line to Table 1 caption:

"*An individual's competence score is the probability that the informant knows (not guesses) the answer to a question, and it is a value between 0 and 1. A group's average estimated competence score above 0.5 indicates moderate agreement in the group, pointing to an underlying model of shared knowledge. Five consensus models were calculated (column 1), for each consensus model the breakdown of mean knowledge scores along group membership is shown for comparison*"

The introduction to the CCA results need a bit more explanation of what exactly your process was and how you tested particinants' knowledge.

How are you demonstrating, or calculating, consensus?

Authors' response:

We explain the consensus analysis process in Methods (3.2) that explains what individual knowledge means in this context, and how consensus is calculated:

"*The consensus model (Romney et al., 1986) estimates shared beliefs relying on three basic steps. First, it uses Principal Component Analysis (PCA) to test whether the responses are consistent with an underlying shared model for the topics covered in the survey. Eigenvalues are calculated to find a shared knowledge-domain, determined by the presence of a single factor that explains most of the variation in the responses, with a first to second eigenvalue ratio greater than, or equal to, 3.0. Secondly, the model provides a measure of individual knowledge for each respondent (a type of 'competence' in the specific shared mental model) by testing each respondent's agreement with shared beliefs via a proportion match matrix that has been corrected for guessing. And finally, it aggregates individual answers to questions by weighting the final cultural model in favour of respondents with high competence. This set of responses produces the consensus-based result, an approximation of the collective knowledge of the group.* "*

And we lay out the boundaries of what the shared knowledge in this case is about:

"*The consensus model can show shared understandings among users of forecasts to reveal patterns of understanding and meaning that impact the adoption of services and products. An advantage of cultural consensus analysis is that a small population of respondents can yield rich observations and data about sector (commercial and recreational) or locality-specific (South African and New Zealand) views and knowledge-domains as they may exist among participants (Weller (2007)).*"

You are referring to knowledge and knowledge questions but do not share those with the readers. Instead you are referring to user needs as "knowledge" but shouldn't this be rather classified as "perceptions or perspectives" seeing that it is highly subjective and will vary from participant to participant with no "right" or "wrong" option. Knowledge can only be tested if there are clear lines between what is a fact and what isn't, and perceptions cannot be evaluated for their "accuracy", nor can they be considered knowledge.

Authors' response:

Consensus analysis approximates the collective knowledge of the group around a particular subject (in our case forecast use/salience). This is referred to as shared knowledge. It is particular to the group being tested and the topic at hand. Right and wrong answers are relative to this commonly held (consensus-based) knowledge. Because the method produces a set of culturally correct answers to the questions, we have a template of right / wrong answers relative to this shared mental model.

Individual knowledge scores are actually called 'competence scores' in cultural consensus analysis. As we write in Results:

"*An individual's competence (or knowledge score) is the probability that the informant knows (not guesses) the answer to a question, and it is a value between 0 and 1.*"

We normally prefer the use of 'knowledge' scores as 'competence' can invite negative reactions especially in culturally sensitive settings. Individual competence scores are related to agreement with others in the group, while the shared knowledge (culturally correct set of answers) is calculated by weighting answers in favour of individuals with high knowledge scores (those who are in high agreement with others). See process description above.

Actions taken:

Because the use of both competence and knowledge scores (they both appear in the paper) seems to cause confusion, we have changed the text to refer to competence scores only.

What heuristics are you referring to (l 315) and what does "some level of consensus" (l 320) actually mean?

Authors' response:

In Table 1 we make a summary conclusion about each of the 5 consensus models. Normally eigen value ratios, mean competence scores are given only and the reader is left to decipher what the results actually mean. This requires familiarity with the method. We used a heuristic in literature that helps to qualify levels of consensus ( see last column Table 1). In the text we explain:

"*We adopt the heuristic by (Caulkins and Hyatt, 1999) to help distinguish varying degrees of consensus, where multiple centres of agreement may exist and form so-called noncoherent models. Where multiple negative competence scores exist, and/or where one subgroup's mean competence is less than .5 (while the other is significantly higher) we identify the model as noncoherent regardless of the eigenvalue ratio. Negative competencies would signal that a participant responded very differently from others.*"

And based on this literature evaluate each of the 5 models as coherent/non-coherent, strong/weak consensus. This is not required in consensus analysis literature, but we deemed it helpful for the readership of Geoscience Communication.

What do the axes on the figures in section 4 depict?  How have you calculated the consensus in Table 2, and why are in some cells the frequencies replaced by green or red ticks?  Shouldn't the frequencies be shown as well?

Authors' response:

Actions taken:

In Multidimensional scaling the axes have no significance beyond indicating relative distance between objects. To clarify this, we added:

"*The x and y axes do not represent meaningful numeric values beyond communicating relative distance between objects.*"

We feel the consensus calculation is sufficiently explained in methods, 3.2. We took no further action here

Table 2 caption explains the presentation of results:

"*Level of consensus measured by the frequency of culturally correct answers (CCA) for all propositions. The whole-group CCA is based on the analysis of the entire dataset consisting of all respondents; this consensus model is shown as true/agreement (green icon) or false/disagreement (red icon) with the propositions. Values are percent of responses matching the whole-group CCA. Where a subgroup's own consensus-model deviates from the whole-group CCA, the added icon shows the correct answer in the sub-group.*"

The green icon means the culturally correct answer is TRUE, red icon means it is FALSE.

The numbers communicate the % of participants in each subgroup who marked the same answer (agreed with) as the shared knowledge, or consensus model of the whole group (all participants). In cases where a subgoup's own consensus model (consensus model run separately, just on the subgroup) produced a different answer than the whole group model, the icon shows this deviating answer.

To help make this clearer, we adjusted the caption to read (changes in bold):

"*Level of consensus measured by the frequency of culturally correct answers (CCA) for all propositions. The whole-group CCA is based on the analysis of the entire dataset consisting of all respondents; the culturally correct answer set (consensus model) is shown as either true/agreement (with a green icon) or false/disagreement (with a red icon) with the propositions. Numeric values are percent of responses matching the whole-group CCA in the relevant subgroups. Where a subgroup's own consensus-model (consensus analysis run separately only with members) deviates from the whole-group CCA, the added icon shows the correct answer in the sub-group.*"

We also adjusted the text in the relevant section (4.2.2) that begins with a description of the Table to start with:

"*Table 2 presents the results of the survey. These are the direct questions and resulting propositions that were distributed in the survey and form the basis of the present study. The column titled "whole-group CCA" is based on the consensus analysis of all respondents together, and it shows the aggregate group belief (culturally-correct answer) with either agreement (green icon) or disagreement (red icon) with the propositions. The other columns indicate the percent frequency of matching answers (or agreement with the whole-group CCA), in each subgroup. In case a subgroup's own consensus-model*

*(consensus analysis run only including its members) produced a group belief that deviates from the whole-group CCA, the added icon indicates the correct answer in the sub-group. "*

In l 403 you refer to "culturally correct answers" - surely this is a typo; if it isn't then I would strongly urge you to revisit some of the cultural geographical literature and reconsider the appropriateness of using such a term.

Authors' response:

In cultural anthropology, specifically in cultural consensus analysis, "culturally correct answer" refers to the shared knowledge approximated in the cultural model of the specific group in question. We are testing if our participants as mariners, as recreational ocean users, form a culture (share an acquired knowledge base used for practical purposes) that is particular to them but we make no assertion about the greater NZ or SA culture. The entire purpose of cultural consensus analysis is to elicit this knowledge domain, that is formed from particular socio-material contexts.

Action taken:

In order to make clear the context in which 'correct' and 'competence' are used in consensus analysis, we added the following paragraph to the end of Methods:

"*Cultural consensus analysis uses 'cultural competence' in very context-specific ways. Culture refers to shared sets of learned knowledge and beliefs among a group of people. Competence is the individual's level of expertise with regard to the set of questions presented, indicating the proportion of items each person knows about the particular domain without moral judgment (Weller 2007). Similarly, the method identifies the 'culturally correct answers' to propositions, from consensus-based results or the most frequently held items of knowledge and belief.*"

You claim that users agreed on "a preferred forecast horizon (3-7 days)" (l 433), but in order to do this you must have provided other options. What were those?

Authors' response:

This question was not added as a poll, but rather as "yes, no" questions. "I don't consider forecasts which exceed a 3-day period to be useful." and users agreed. These results were also given in Table 2.

All in all, section 4 leaves quite a few questions for me, which I hope you can address in a revision of the MS.

Similarly, the discussion and conclusion would benefit from a more critical examination of what you have actually done in your research (compared information obtained from two countries with relatively small samples) vs. what you claim (cross-sectoral and cross-cultural research). What you have done is an important step in the right direction

and deserves to be published but I think it is naive to make it sound as 'grander' than it was.

Authors' response:

We feel that we have sufficiently shown that our method is robust to back claims that we in fact explore cultural variations among participants as a factor of sectoral traits (recreational and commercial) and geographical traits (SA and NZ).

You refer to the users having had "a lot of experience with coastal and ocean activities..." (l 472) but it is not clear how and why one could arrive at this conclusion from the data you collected. What do you mean by "their interpolation" (l 479) or "users' decision quality" (l 530) - these terms will have to be defined more clearly.

We stated that users are experienced as large amounts of people had 10 years' or more experience with these platforms (refer to Figure 1 E).

The sentence regarding the interpolation and decision quality has been reworded:

Their interpolation (of wave conditions from the offshore to the nearshore) also exceeds most high-resolution models and (mostly) unknowingly compensate for various coastal processes (like friction, refraction, shoaling etc.). The same reasoning applies to most commercial users (including Search and Rescue operators).

Decision quality is defined as the users' ability to make informed decisions, correctly. Thus, the user is empowered to make the correct decision

Overall, I think you have collected some interesting data that deserve to be published, and I am sure that you will be able to carefully review and revise your MS to bring it into a publishable form. I wish you all the best on this journey and look forward to seeing your work in the public realm.

Thank you very much for taking the time to help us make our contribution even better.

---

## Author Response (AR2)

Thank you very much! All corrections have been made.